# Loss of p300 in proximal tubular cells reduces renal fibrosis and endothelial-mesenchymal transition

Hyunsik Kim [1], Soo-Yeon Park [1], Soo Yeon Lee [1], Jae-Hwan Kwon[1], Seunghee Byun [1],

Byounghwi Ko[2], Jung-Yoon Yoo[3], Beom Seok Kim [2✉], Beom Jin Lim [4✉] & Ho-Geun Yoon [1✉]

## Abstract

**Chronic kidney disease (CKD) has a high prevalence worldwide and is typically accompanied by severe fibrosis. However, the exact pathogenesis of renal fibrosis and effective treatments have yet to be identified. In this study, we found that expression of the histone-acetyltransferase p300 was increased in focal segmental glomerulosclerosis patients and several distinct mouse models of renal fibrosis. Moreover, we showed that the AKT-mediated phosphorylation of Ser-1834 of p300 increased the stability of p300 upon renal fibrosis induction, and conversely, PPM1K specifically dephosphorylated p300 at Ser-1834, resulting in a significant reduction in p300 stability and renal fibrosis. Interestingly, increased p300 in proximal tubular cells (PTCs) promoted renal fibrosis development by mediating the endothelial to mesenchymal transition (EndMT) via upregulation of the mesenchymal-transition-related secreted proteins POSTN, FSTL1, and FSCN1. Both EndMT and renal fibrosis were significantly diminished by either PTC-specific deletion of p300 gene or selective inhibitors of p300. Collectively, our results demonstrate the role of p300 in the development of renal fibrosis, and suggest that p300 is a promising target for treatment of advanced CKD.**

**Keywords** Chronic Kidney Disease; Endothelial-mesenchymal Transition; p300; Renal Fibrosis, Proximal Tubular Cells
**Subject Category** Urogenital System

## Introduction

Chronic kidney disease (CKD) is a collective term for chronic conditions characterized by a persistent decrease in glomerular filtration rate due to various etiologies (Adamczak and Surma, 2021; Anders et al, 2018; de Cos et al, 2022; Devarajan et al, 2022; Flagg, 2018; Hamrahian and Falkner, 2017; Reiss et al, 2015; Vehaskari, 2011). CKD is increasing in prevalence worldwide, affecting over 10% of the global population, and has emerged as a significant social problem (Kovesdy, 2022; Lv and Zhang, 2019). In patients with end-stage renal failure due to CKD, dialysis and transplantation are the only treatment options (Wilkinson et al, 2020).

Fibrosis, a major consequence of CKD progression, is an irreversible process induced by the deposition of extracellular matrix (ECM) (Humphreys, 2018; Li et al, 2022; Panizo et al, 2021). ECM replaces normal tissue, which leads to functional failure of organs (Wynn, 2008). Fibrosis is regulated by various physiological pathways and mediators such as cytokines, chemokines, gut microbiota, physical stretching of the kidneys, and fatty acid oxidation (Borthwick, 2016; Humphreys, 2018; Kang et al, 2015; Li et al, 2022; Meng et al, 2016; Wynn, 2008; Zhang et al, 2021; Zhao et al, 2022; Zhou et al, 2022). Renal fibrosis is accompanied by inflammation, activation of resident fibroblasts, the epithelial-to-mesenchymal transition (EMT), endothelial-to-mesenchymal transition (EndMT), and rarefaction of the peritubular capillary network, leading to malfunction of the nephron system (Ballermann and Obeidat, 2014; Lian et al, 2011; Liu and Wang, 2022; Maremonti et al, 2022; Sanz et al, 2023; Yuan et al, 2019; Zhang et al, 2023). Fibrosis is driven by deposition of ECM secreted from myofibroblasts, thus tracing the origin of myofibroblasts is important for understanding fibrotic pathogenesis (Wynn and Ramalingam, 2012). Myofibroblasts involved in renal fibrosis originate from a variety of cells, including fibroblasts, fibrocytes, macrophages, pericytes, epithelial cells, and endothelial cells (Grgic et al, 2012; LeBleu et al, 2013; Loeffler and Wolf, 2015; Sun et al, 2016; Zeisberg et al, 2008). It is well established that resident fibroblasts are the main progenitors of myofibroblasts in fibrosis progression. However, other cells can also transition to myofibroblasts. In particular, the main kidney cell types of epithelial and endothelial cells can become myofibroblasts in renal fibrosis (Carew et al, 2012; Li et al, 2019; Yuan et al, 2019; Zeisberg et al, 2008). Although many studies have reported that the EMT plays an important role in the development of fibrosis, recent renal fibrosis studies have reported that the proportion of epithelial cells that transition to mesenchymal cells is not large, but that the proportion of endothelial cells that transition to mesenchymal cells is larger than that observed in the EMT (LeBleu et al, 2013; Loeffler and Wolf, 2015). Therefore, additional studies are needed to unravel the

---

[1]Department of Biochemistry and Molecular Biology, Severance Medical Research Institute, Brain Korea 21 PLUS Project for Medical Sciences, Yonsei University College of Medicine, Seoul 03722, Korea. [2]Department of Internal Medicine, Yonsei University College of Medicine, Seoul 03722, Korea. [3]Department of Biomedical Laboratory Science, Yonsei University MIRAE Campus, Wonju 26493, Korea. [4]Department of Pathology, Yonsei University College of Medicine, Seoul 03722, Korea. ✉E-mail: docbsk@yuhs.ac; bjlim@yuhs.ac; YHGEUN@yuhs.ac

molecular mechanisms that regulate EndMT and to determine the detailed role of EndMT in the development of renal fibrosis.

Epigenetic dysregulation through DNA methylation, histone acetylation, and/or histone methylation has been implicated in a variety of diseases including fibrosis of diverse organs (Hillyar et al, 2020; Sarkar et al, 2013; Surace and Hedrich, 2019; Timmermann et al, 2001; Xue et al, 2021; Zhang et al, 2020). p300, also known as histone acetyltransferase EP300 or E1A-associated protein p300, acetylates lysine residues on histones, resulting in changes in gene transcription (Dekker and Haisma, 2009; Ghosh, 2014; Giles et al, 1998; Iyer et al, 2004). p300 in particular has been reported to be involved in fibrosis in several organs (Gao et al, 2021b; Ghosh et al, 2013; Ghosh and Varga, 2007; Ghosh and Vaughan, 2012; Lee et al, 2023; Rai et al, 2019; Rubio et al, 2023). In liver fibrosis, p300 produced by hepatic endothelial cells regulates fibrosis by mediating C-C motif chemokine ligand 2 while in hepatic stellate cells, p300 protein stability is increased by mechanical stress, which promotes their transition into myofibroblasts (Dou et al, 2018; Gao et al, 2021b). Elevation of p300 has been reported in idiopathic pulmonary fibrosis (IPF) patients, and p300 is specifically upregulated in alveolar type-2 cells, which regulate fibrosis by promoting M2 macrophage proliferation (Lee et al, 2023). p300 has been reported to have multiple functions in renal tubular cells during renal disease development. In tubular cells, p300 over-expression promotes the acetylation of STAT3, leading to increased expression of collagen and fibronectin. In addition, increased p300 expression in patients with diabetic nephropathy (DN) has been reported (Gong et al, 2022; Ni et al, 2014). In a human proximal tubule cell line mimicking DN, p300 level was increased, which promoted EMT progression and increased ECM protein expression (Gong et al, 2022). However, the pathophysiological roles of p300 in the development of renal fibrosis require further clarification.

In this study, we found increased expression of p300 in PTCs, focal segmental glomerulosclerosis (FSGS) patients, and mouse models of renal fibrosis. We investigated the PTC-specific function of p300 and the mechanism underlying this protein's contribution to the progression of renal fibrosis. Our findings collectively demonstrate the functional significance of p300 in the development of renal fibrosis and suggest that p300 is a promising therapeutic target in advanced CKD.

# Results

## PTC-specific deletion of p300 suppresses the development of renal fibrosis

To investigate the clinical relevance of p300 in CKD with fibrosis, we first examined the expression of p300 protein in renal tissues of idiopathic FSGS patients with fibrosis and those with minimal change disease (MCD) without fibrosis seen at Yonsei University Severance Hospital. We observed significantly higher expression of p300 in the FSGS group than the MCD group and normal kidneys (Fig. 1A). Next, to further verify the results obtained in clinical samples, we examined the expression of p300 in multiple renal fibrosis mouse models including a unilateral ureteral obstruction (UUO) model, folic acid (FA) model, streptozotocin-uninephrectomy (STZ-UNx) model. A significant increase in the expression of p300 was observed in kidney tissues from all three renal fibrosis mouse models (Fig. 1B; Appendix Fig. S1A–F), but

not in other histone acetyltransferases (Appendix Fig. S2). It is noteworthy that we observed elevated p300 protein levels during fibrosis development, particularly in renal epithelial cells. Next, to examine the cell type-specific expression of p300 in kidney tissues from renal fibrosis mouse models, we performed serial immuno-histochemistry (IHC) staining using antibodies against proximal tubules (aquaporin-1, AQP1), collecting ducts (aquaporin-2, AQP2), glomeruli (Wilms tumor protein, WT1), and blood vessels (PECAM1, CD31), and evaluated changes in p300 expression in each cell type by H-scoring. We observed a significant increase in p300 expression in AQP1-positive cells and further confirmed, using co-immunofluorescence (co-IF), that p300 expression was significantly elevated in AQP1-positive epithelial cells, a marker for proximal tubule cells, but not in AQP2-positive cells (Fig. 1C,D; Appendix Fig. S3A–E). To further investigate the correlation between p300 expression in PTCs and fibrosis progression, we assessed p300 expression in kidney tissues from mice sacrificed at 2, 4, 6, 8, and 14 days after UUO surgery using H-scoring. p300 protein expression in PTCs increased time-dependently with renal fibrosis progression observed until 8 days after surgery (Fig. 1E; Appendix Fig. S4A). Similar expression patterns of p300 were observed in the other renal fibrosis-induced mouse models, demonstrating that p300 expression is significantly upregulated in PTCs during renal fibrosis (Appendix Fig. S4B).

To investigate the physiological roles of p300 in PTCs during renal fibrosis, we generated conditional knockout (cKO) mice by specific genetic deletion of p300 in PTCs using γGT-1-Cre and p300 floxed mice (Appendix Fig. S5A). We verified the knockout of p300 using tissue IHC in kidney sections, and co-IF in isolated PTCs from mouse kidneys (Appendix Fig. S5B–E). Using p300 cKO and wild-type (WT) mice, we performed UUO surgery and evaluated fibrotic changes 8 days after surgery. Fibrosis progression was significantly reduced in p300 cKO mice compared to WT mice, as confirmed by Masson trichrome staining (MTS), Sirius red staining, and α-smooth muscle actin immunostaining (Fig. 1F; Appendix Fig. S6A,B). We also observed a significant reduction in the expression of fibrosis-related genes and proteins in renal tissues from p300 cKO mice (Fig. 1G; Appendix Fig. S6C). Moreover, kidney injury marker-1 (KIM1) levels were also decreased in p300 cKO mice compared to WT mice (Appendix Fig. S6D). Similar results were observed in FA-induced renal fibrosis mouse models, and renal function markers (blood urea nitrogen and serum creatinine) were also alleviated (Figs. 1H,I and EV1A,B). To further validate the pro-fibrotic roles of proximal tubular p300 in FSGS patients, we utilized the Adriamycin-induced kidney injury model, a representative animal model that mimics FSGS. Consistent with other models, p300 expression was elevated in Adriamycin-induced mouse kidney tissues, and fibrosis progression was significantly reduced in p300 cKO mice compared to WT mice following Adriamycin treatment (Appendix Fig. S7A–E). These results suggest that PTC-specific p300 plays a critical physiological role in the progression of renal fibrosis.

## TGFβ activation increases the AKT-mediated phosphorylation of p300 at Ser-1834 by dissociating PPM1K phosphatase from p300

To elucidate the molecular mechanism of renal fibrosis mediated by PTC-specific p300, we first treated primary renal proximal tubular

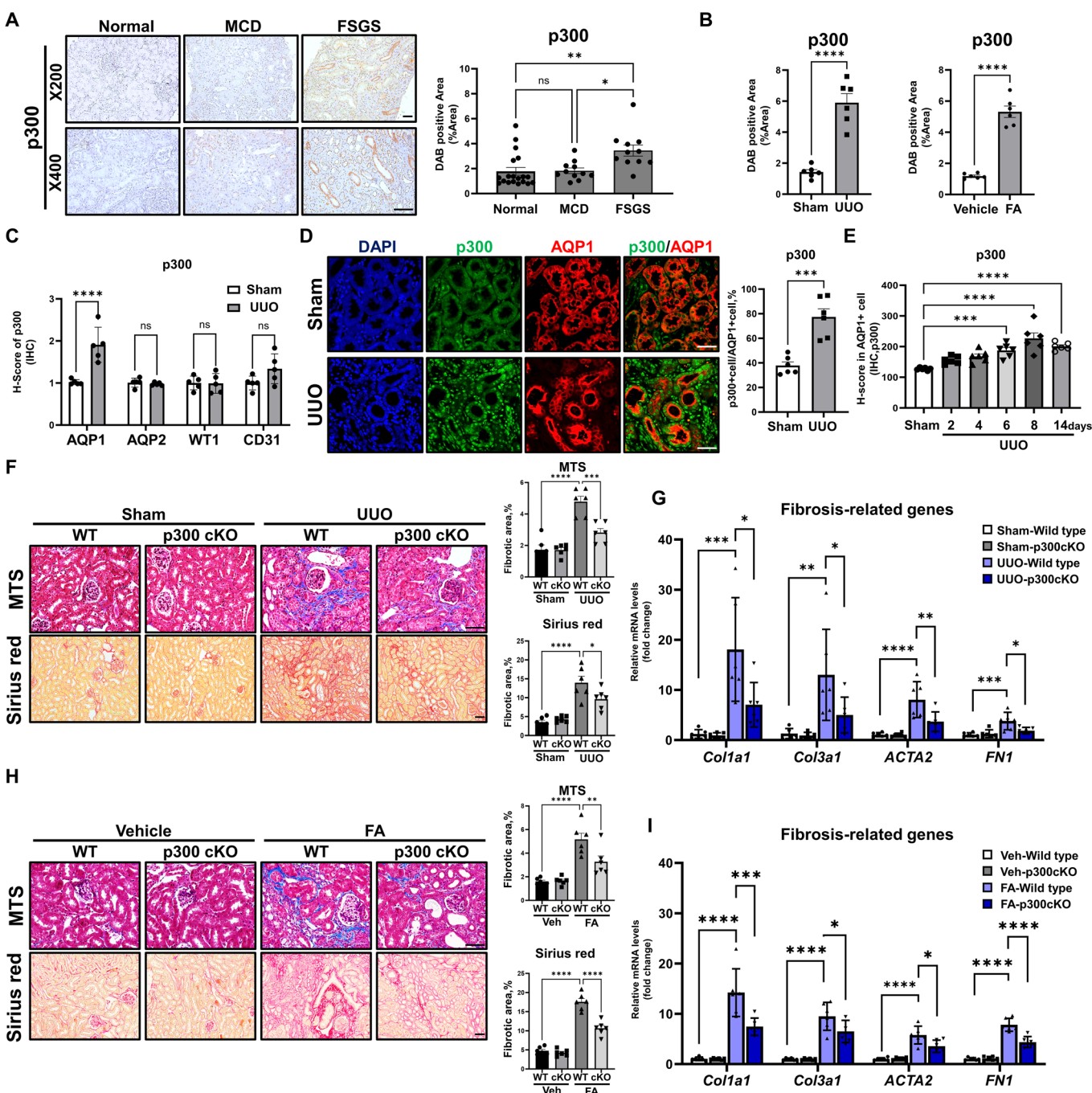

epithelial cells (PTECs) with transforming growth factor-β (TGFβ), a well-known fibrotic stimulus, and observed a time-dependent increase in p300 protein levels (Appendix Fig. S8A). However, mRNA levels of p300 were not altered upon TGFβ treatment (Appendix Fig. S8B). We also observed a similar result using the human renal proximal tubular cell line HK2 (Fig. 2A,B; Appendix Fig. S8C). Recent studies in IPF and non-alcoholic steatohepatitis demonstrated that p300 protein levels are regulated through protein stabilization, rather than mRNA (Dou et al, 2018; Lee et al, 2023). Thus, we next investigated whether p300 protein stability is regulated by TGFβ in HK2 cells. After treatment with

MG132 (proteasome inhibitor) and TGFβ, we observed that the accumulation of p300 induced by MG132 treatment was further enhanced by co-treatment with TGFβ, confirming that TGFβ promotes the stabilization of p300 (Fig. 2C; Appendix Fig. S8D). The stability of p300 has been reported to be regulated by phosphorylation at Serine-1834 of p300 through the AKT signaling pathway (Dou et al, 2018; Huang and Chen, 2005). Accordingly, we verified in HK2 cells that TGFβ stimulation induces a time-dependent increase in p300 phosphorylation at Ser-1834 (Appendix Fig. S9A). In addition, TGFβ stimulation in HK2 cells led to AKT phosphorylation, which subsequently promoted p300

**Figure 1. Elevation of p300 in proximal tubule cells promotes renal fibrosis.**

(A) Representative image of p300 immunohistochemistry (IHC) in kidneys from minimal change disease (MCD, $n = 11$), focal segmental glomerulosclerosis (FSGS, $n = 11$), and normal kidneys ($n = 19$). The graph represents the quantification of the DAB-positive area in histological staining images. Bar = 100 μm. Normal vs FSGS, $P = 0.0036$; MCD vs FSGS, $P = 0.0127$. (B) Quantification of p300 IHC images in kidney tissues from UUO- and FA-induced mouse fibrosis models ($n = 6$ per group). Sham vs UUO, $P < 0.0001$; Vehicle vs FA, $P < 0.0001$. (C) p300 expression within AQP1-, AQP2-, WT1-, and CD31-positive cells from the UUO-induced mouse fibrosis model analyzed by H-scoring ($n = 5$ per group). Sham vs UUO (AQP1): $P < 0.0001$. (D) Representative images of p300 and AQP1 co-immunofluorescence (IF) in mouse kidney tissue from the UUO-induced mouse fibrosis model ($n = 6$ per group). The graph represents the proportion of p300-positive cells among AQP1-positive cells. Bar = 100 μm. Sham vs UUO, $P = 0.0002$. (E) Expression of p300 in AQP1-positive cells from the UUO-induced mouse fibrosis model analyzed by H-scoring ($n = 6$ per group). Sham vs 6 days, $P = 0.001$; Sham vs 8 days, $P < 0.0001$; Sham vs 14 days, $P < 0.0001$. (F) Representative image of Masson trichrome staining (MTS) and Sirius red staining of kidney tissues from wild-type and p300 knockout (cKO) UUO-induced mouse fibrosis models. The graph represents the quantification of fibrotic areas in histological staining images models ($n = 6$ per group). Bar = 100 μm. MTS: WT-Sham vs WT-UUO, $P < 0.0001$; WT-UUO vs cKO-UUO, $P = 0.0003$. Sirius red: WT-Sham vs WT-UUO, $P < 0.0001$; WT-UUO vs cKO-UUO, $P = 0.0423$. (G) mRNA levels of fibrosis-related genes in kidney tissues from wild-type and p300 knockout (cKO) UUO-induced mouse fibrosis models ($n = 6$ per group). *Col1a1*: WT-Sham vs WT-UUO, $P = 0.0003$; WT-UUO vs cKO-UUO, $P = 0.0145$. *Col3a1*: WT-Sham vs WT-UUO, $P = 0.0027$; WT-UUO vs cKO-UUO, $P = 0.0475$. *ACTA2*: WT-Sham vs WT-UUO, $P < 0.0001$; WT-UUO vs cKO-UUO, $P = 0.0072$. *FN1*: WT-Sham vs WT-UUO, $P = 0.0009$; WT-UUO vs cKO-UUO, $P = 0.0197$. (H) Representative image of Masson trichrome staining (MTS) and Sirius red staining of kidney tissues from wild-type and p300 knockout (cKO) FA-induced fibrosis mouse models. The graph represents the quantification of fibrotic areas in histological staining images models ($n = 6$ per group). Bar = 100 μm. MTS: WT-Sham vs WT-UUO, $P < 0.0001$; WT-UUO vs cKO-UUO, $P = 0.009$. Sirius red: WT-Sham vs WT-UUO, $P < 0.0001$; WT-UUO vs cKO-UUO, $P < 0.0001$. (I) mRNA levels of fibrosis-related genes in kidney tissues from wild-type and p300 knockout (cKO) FA-induced fibrosis mouse models ($n = 6$ per group). *Col1a1*: WT-Sham vs WT-UUO, $P < 0.0001$; WT-UUO vs cKO-UUO, $P = 0.0008$. *Col3a1*: WT-Sham vs WT-UUO, $P < 0.0001$; WT-UUO vs cKO-UUO, $P = 0.0379$. *ACTA2*: WT-Sham vs WT-UUO, $P < 0.0001$; WT-UUO vs cKO-UUO, $P = 0.0106$. *FN1*: WT-Sham vs WT-UUO, $P < 0.0001$; WT-UUO vs cKO-UUO, $P < 0.0001$. Data are presented as mean ± SEM. ns = not significant, \*$P < 0.05$, \*\*$P < 0.01$, \*\*\*$P < 0.001$, and \*\*\*\*$P < 0.0001$ by t-test (for two-group comparisons) and ordinary one-way ANOVA (for multiple groups). Source data are available online for this figure.

phosphorylation (Appendix Fig. S9B). Conversely, blocking either the TGFβ or AKT signaling pathway led to a decrease in both p300 phosphorylation and p300 protein levels (Fig. 2D). To elucidate the function of Ser-1834 of p300 during stability regulation, we generated a mutant construct where Ser-1834 was replaced by alanine (S1834A). In HK2 cells, TGFβ stimulation increased phosphorylation levels of wild-type p300 but not mutant p300 (S1834A) (Appendix Fig. S9C). A cycloheximide (CHX) chase assay revealed reduced protein stability of the S1834A mutant of p300 compared to wild-type p300 (Appendix Fig. S9D). To determine whether AKT signaling stabilizes p300, we performed a CHX chase assay with or without AKT inhibitor (LY294002). The results showed that the TGFβ-induced increase in p300 stability was reduced upon AKT inhibitor treatment compared to TGFβ treatment alone (Fig. 2E). Furthermore, we confirmed an increase in phosphorylation of p300 at Ser-1834 in our UUO-induced renal fibrosis mouse model by IHC and IF staining (Figs. 2F and EV2A,B). Together, these findings indicate that AKT-mediated phosphorylation of p300 at Ser-1834 is necessary for stabilization of p300 upon renal fibrosis induction.

Protein phosphorylation is regulated by phosphatases in a reversible reaction. To identify the partner phosphatase of p300 that de-phosphorylates Ser-1834, we screened multiple phosphatases and identified binding of PPP6C, PPM1D, PPM1M, PPM1K, and PPEF1 to p300 through in vitro transcription and immunoprecipitation assays (Appendix Fig. S10A). We found that PPM1K, among the phosphatases that bound to p300, significantly reduced p300 protein levels and phosphorylation of p300 in HK2 cells (Fig. 2G,H; Appendix Fig. S10B). Moreover, overexpression of the inactive mutant PPM1K N94K failed to reduce p300 protein levels and phosphorylation upon TGFβ treatment, compared to wild-type PPM1K (Appendix Fig. S10C). To examine whether TGFβ treatment affects the interaction between p300 and PPM1K, we performed co-immunoprecipitation (Co-IP) analysis in HK2 cells overexpressing HA-tagged p300 and Flag-tagged PPM1K. This analysis revealed that TGFβ treatment diminished the interaction between PPM1K and p300 (Appendix Fig. S11A). Similar results

were observed using proximity ligation assays (PLA) in kidneys from UUO-induced renal fibrosis mouse models and TGFβ-stimulated HK2 cells (Fig. 2I; Appendix Fig. S11B). Taken together, our findings demonstrate that TGFβ signaling increases AKT-mediated phosphorylation and stabilization of p300 by disrupting PPM1K-p300 interaction during renal fibrosis progression.

## PPM1K negatively regulates renal fibrosis via inhibition of p300 stability

Based on our finding that PPM1K inhibits AKT-mediated p300 phosphorylation and reduces p300 stability, we aimed to explore its clinical relevance to the development of renal fibrosis. To address this, we first analyzed a Gene Expression Omnibus (GEO) dataset (GSE142025) revealed a significant reduction in PPM1K expression in CKD patients (Appendix Fig. S12A). Similarly, single-cell RNA sequencing analysis of CKD patients (GSE183279) revealed decreased PPM1K expression levels and a reduced proportion of PPM1K-expressing PTCs (Appendix Fig. S12B).

To further investigate these findings, we examined PPM1K expression in kidney tissues from CKD patients in the Yonsei University Severance Hospital cohort and observed significantly lower PPM1K expression in FSGS patients with fibrosis compared to MCD patients without fibrosis and normal kidneys (Fig. 3A). Correlation analysis ($R^2 = 0.4034$, $p < 0.0001$) and IHC staining of p300 and PPM1K in the same tissue regions indicated an inverse correlation between p300 and PPM1K expression in CKD patients (Figs. 3B and S12C). We also found that, in contrast to the p300 expression, PPM1K decreases during fibrosis progression in the UUO-induced renal fibrosis mouse model (Figs. 3C,D and S12D,E). Consistently, correlation analysis of p300 and PPM1K expression in the UUO-induced renal fibrosis mouse model revealed an inverse correlation between p300 and PPM1K expression, similar to that observed in CKD patients (Appendix Fig. S12F). Furthermore, we verified that, in contrast to the upregulation of p300, PPM1K expression was downregulated in PTECs and HK2 cells in response to TGFβ treatment (Fig. 3E; Appendix Fig. S12H). Taken together,

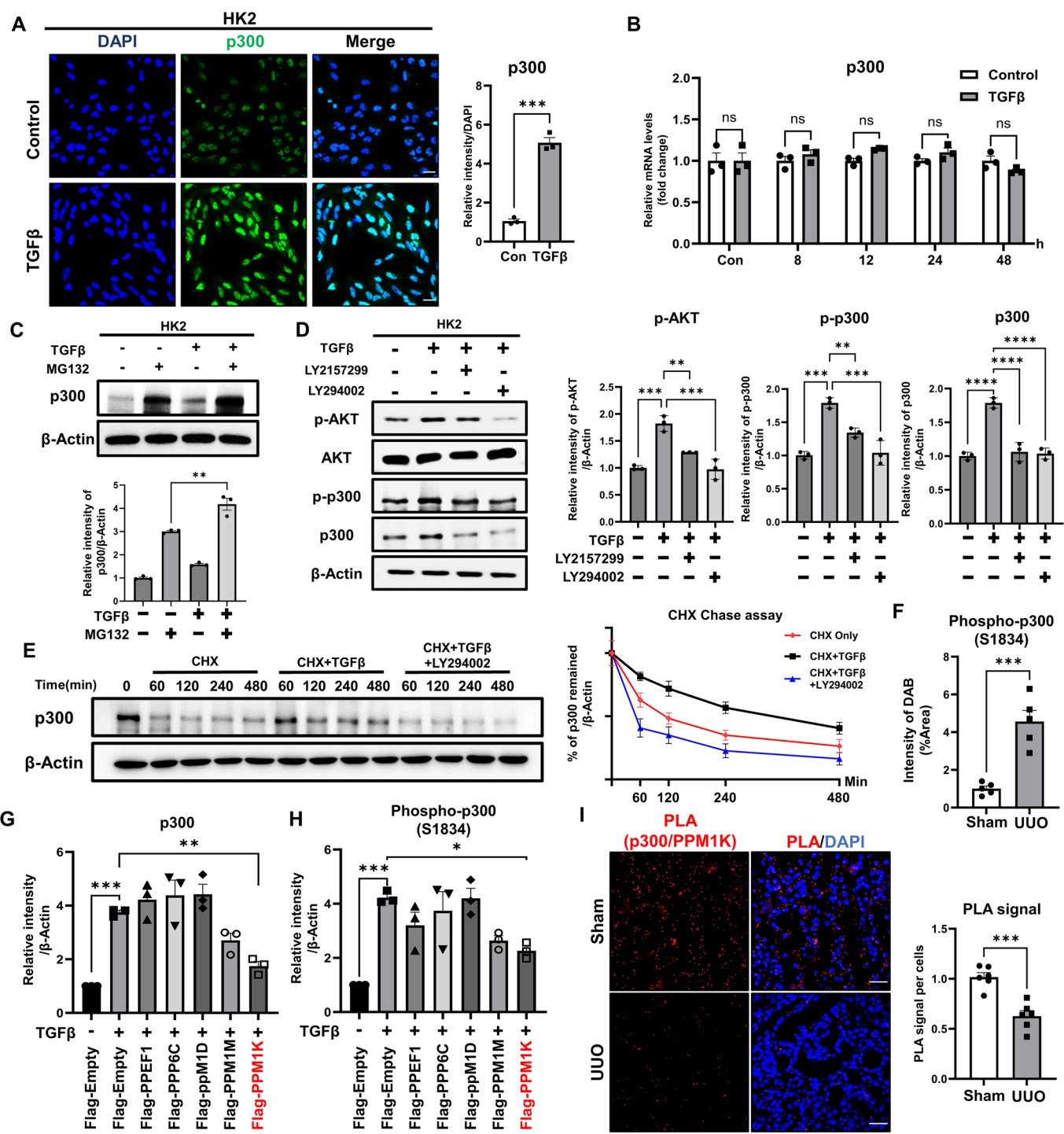

these results demonstrate that PPM1K expression is inversely correlated with p300 expression and renal fibrosis progression, providing insight into its potential role in renal pathophysiology.

To unravel the physiological roles of PPM1K in renal fibrosis, we analyzed the RNA-sequencing dataset of a PPM1K-overexpressing HK2 cell line (GSE212681) and identified reduced expression of fibrosis-related genes such as *Col1A1*, *Col2A1*, *CTGF*, *vimentin*, and *FN1* (Fig. EV3). Furthermore, overexpression of wild-type PPM1K significantly suppressed the TGFβ-mediated

increase in fibrosis-related target expression at both the protein and mRNA levels compared to inactive PPM1K mutant (Appendix Fig. S13A,B). Conversely, knockdown of PPM1K further promoted the TGFβ-mediated increase in fibrosis-related target expression in HK2 cells (Appendix Fig. S13C,D).

We next examined the roles of PPM1K in renal fibrosis development in vivo. To overexpress PPM1K in vivo, we generated adenovirus serotype 5 (Ad5) vectors carrying either wild-type PPM1K or mutant PPM1K N94K tagged with GFP, which were

◄ **Figure 2.** The phosphatase PPM1K reduces p300 stability via de-phosphorylation of p300 at Ser-1834 in proximal tubule cells.

(A) Representative image of p300 immunofluorescence (IF) in HK2 cells treated with TGFβ. The graph represents quantification of p300 immunofluorescence, normalized to DAPI ($n = 3$ per group). Bar = 25 μm. Control vs TGFβ, $P = 0.0001$. (B) mRNA levels of p300 in HK2 cells treated with TGFβ for the indicated time ($n = 3$ per group). Independent control samples were used at each time point, and data were normalized to the corresponding control. (C) Protein levels of p300 in HK2 cells treated with TGFβ and MG132 for 6 h. The graph represents quantification of intensity in western blot images, normalized to β-Actin ($n = 3$ per group). MG132 vs MG132 + TGFβ, $P = 0.0013$. (D) Protein levels of AKT, p-AKT, p300, and p-p300 in HK2 cells treated with TGFβ, LY2157299 (TGFβ inhibitor) and LY294002 (AKT inhibitor) for 6 h. The graph represents quantification of intensity in western blot images, normalized to β-Actin ($n = 3$ per group). p-AKT: Control vs TGFβ, $P = 0.0001$; TGFβ vs TGFβ + LY2157299, $P = 0.0026$; TGFβ vs TGFβ + LY294002, $P = 0.0001$. p-p300: Control vs TGFβ, $P = 0.0001$; TGFβ vs TGFβ + LY2157299, $P = 0.0049$; TGFβ vs TGFβ + LY294002, $P = 0.0002$. p300: Control vs TGFβ, $P < 0.0001$; TGFβ vs TGFβ + LY2157299, $P < 0.0001$; TGFβ vs TGFβ + LY294002, $P < 0.0001$. (E) Protein level of p300 in HK2 cells treated with TGFβ, CHX, and LY294002 (AKT inhibitor) for the indicated times. The graph represents quantification of p300 intensity in western blot images ($n = 3$ per group). (F) Quantification of phosphorylated-p300 in kidney tissue from the UUO-induced mouse fibrosis model ($n = 6$ per group). Sham vs UUO, $P = 0.0004$. (G, H) Protein levels of p300 and p300 phosphorylated at Serine 1834 in HK2 cells transfected with phosphatases and treated with TGFβ. The graph represents quantification of p300 and p300 phosphorylated at Serine 1834 based on western blot image, normalized to β-Actin ($n = 3$ per group). p300: Control vs TGFβ, $P = 0.0005$; Control vs TGFβ + PPM1K, $P = 0.0085$. p-p300: Control vs TGFβ, $P = 0.0004$; Control vs TGFβ + PPM1K, $P = 0.0277$. (I) Representative image of the proximity ligation assay (PLA) in mouse kidney tissues from the UUO-induced mouse fibrosis model ($n = 6$ per group). The graph represents quantification of PLA signal, normalized to DAPI. Bar = 100 μm. Sham vs UUO, $P = 0.0003$. Data are presented as mean ± SEM. ns = not significant, *$P < 0.05$, **$P < 0.01$, ***$P < 0.001$, and ****$P < 0.0001$ by t-test (for two-group comparisons) and ordinary one-way ANOVA (for multiple groups). Source data are available online for this figure.

injected into mice via the subcapsular route. The viral delivery was validated by confirming GFP expression (Appendix Fig. S14). Overexpression of PPM1K via subcapsular injection of Ad5-PPM1K into UUO kidneys significantly reduced p300 and phospho-p300 expression in kidney tissues compared to the inactive Ad5-PPM1K N94K mutant (Appendix Fig. S15A,B). Moreover, we observed that PPM1K overexpression attenuated fibrosis and reduced mRNA levels of fibrosis-related genes in a UUO-induced renal fibrosis mouse model (Fig. 3F,G). Collectively, these data indicate that PPM1K suppresses the development of renal fibrosis via inhibition of p300 stability and function.

## PTC-specific p300 mediates the endothelial-mesenchymal transition in renal fibrosis development

Given the physiological importance of p300 in renal fibrosis, we sought to determine the molecular mechanisms by which PTC-specific p300 mediates the progression of renal fibrosis. To identify genes transcriptionally regulated by p300, we first performed RNA-sequencing using kidney tissue from p300 cKO mice and CUT&TAG sequencing using primary kidney proximal tubule cells isolated from p300 cKO mice. RNA-seq analysis revealed 702 genes that were downregulated in p300 cKO mice with UUO compared to WT mice with UUO (fold-change > 1.5, 2Log2). Moreover, comparative analysis of CUT&TAG sequencing data between WT and p300 cKO mice, both subjected to UUO, identified more than 800 genes regulated by PTC-specific p300 upon induction of renal fibrosis. Then, we identified 28 commonly regulated genes in both RNA-sequencing and CUT&TAG-sequencing analyses (Fig. 4A,B). To investigate the biological pathways regulated by PTC-specific p300 during the development of renal fibrosis, we performed GO enrichment analysis using PANTHER with p300 cKO mice. As a result, we identified EMT and angiogenesis as the major pathways regulated by PTC-specific p300. Furthermore, gene set enrichment analysis (GSEA) revealed significant changes in the expression of EMT-related genes specifically regulated by PTC-specific p300 (Fig. 4C; Appendix Fig. S16A,B). Notably, analysis of kidney single-cell RNA sequencing data from CKD patients revealed a significant increase in mesenchymal marker levels, not only in PTCs but also prominently in endothelial cells (Appendix Fig. S17A,B). Therefore, we focused on EndMT, which influences angiogenesis and shows similar gene expression patterns to EMT.

Among the genes predicted to be regulated by p300 in proximal tubule cells during renal fibrosis, we selected the *POSTN*, *FSTL1*, and *FSCN1* genes, as these encode secreted proteins that mediate interactions with surrounding cells and affect the mesenchymal transition (Cho et al, 2023; Dorafshan et al, 2022; Jia et al, 2021; Jin et al, 2020; Li et al, 2018; Liu et al, 2017; Louis et al, 2019; Pontemezzo et al, 2021; Sisto et al, 2022; Wu et al, 2024; Xu-Dubois et al, 2020; Xu-Dubois et al, 2016; Zhang et al, 2018). ChIP-sequencing analysis of human fetal kidney cortex tissue using an antibody against p300 (GSE75948) showed enrichment of p300 near the transcription start site (TSS) of the *POSTN*, *FSTL1*, and *FSCN1* genes (Appendix Fig. S18A). ATAC-sequencing analysis of PTCs from an ischemia-reperfusion injury-induced renal fibrosis mouse model (GSE197815) revealed increased chromatin accessibility in the promoter regions of *POSTN*, *FSTL1*, and *FSCN1* genes during the progression of renal fibrosis (Appendix Fig. S18B). In addition, by analyzing a public GEO dataset of CKD patients, we found that the expression of these three genes was increased in CKD patients and in PTCs from CKD patients, suggesting that increased p300 upon induction of renal fibrosis activates transcription of *POSTN*, *FSTL1*, and *FSCN1* genes (Appendix Fig. S19A,B).

Next, we assessed levels of POSTN, FSTL1, and FSCN1 proteins in HK2 culture media after TGFβ treatment. Both mRNA and protein levels of these three targets increased upon TGFβ treatment in a time-dependent manner as p300 level increased (Appendix Fig. S20A,B). Genetic deletion of p300 in HK2 cells suppressed the TGFβ-induced upregulation of mRNA expression and secretion of these three proteins (Fig. 4D; Appendix Fig. S20C). Overexpression of p300 in HK2 cells further increased TGFβ-induced transcription of these three genes (Appendix Fig. S20D). Moreover, the increased expression of these three proteins in the UUO mouse model was reversed in the kidneys of p300 cKO mice (Fig. 4F; Appendix Fig. S20E). Importantly, overexpression of PPM1K, but not mutant PPM1K N94K, suppressed the expression of the three genes in both kidney tissues of UUO-induced renal fibrosis model mice and HK2 cells (Fig. 4G; Appendix Fig. S20F).

We next investigated whether PTC-specific p300 mediates the EndMT through secretion of these three proteins during the development of renal fibrosis. For this, we first examined quantitative changes in endothelial cells by IHC using the endothelial cell marker, CD31, in UUO-induced p300 cKO mice kidneys. The number of endothelial cells in the kidney tissues decreased in UUO-induced WT

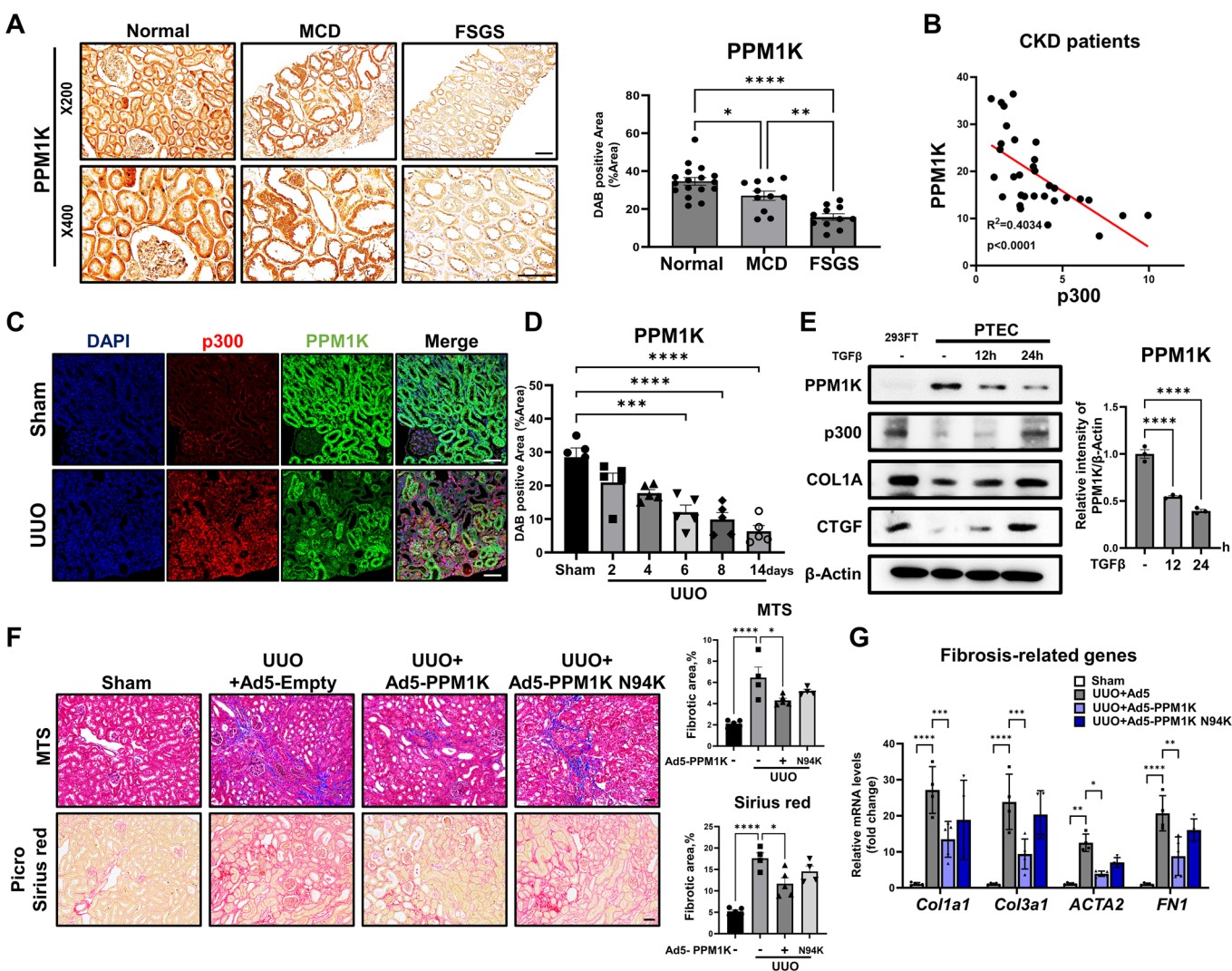

**Figure 3. PPM1K is a negative regulator of renal fibrosis.**

(A) Representative image of PPM1K immunohistochemistry (IHC) in kidney tissue from minimal change disease (MCD, $n = 11$), focal segmental glomerulosclerosis (FSGS, $n = 11$), and normal kidneys ($n = 17$). The graph represents quantification of the DAB-positive area in histological staining images. Bar $= 100\,\mu m$. Normal vs MCD, $P = 0.0367$; Normal vs FSGS, $P < 0.0001$; MCD vs FSGS, $P = 0.0039$. (B) Correlation analysis of p300 and PPM1K expression in immunohistochemistry (IHC) images of kidneys from CKD patients (MCD, FSGS, and IgAN) ($n = 32$). (C) Representative image of p300 and PPM1K coimmunofluorescence (IF) in mouse kidney tissues from the UUO-induced mouse fibrosis model. Bar $= 100\,\mu m$. (D) The graph represents quantification of PPM1K immunohistochemistry (IHC) images in kidney tissue from the UUO-induced mouse fibrosis model ($n = 5$ per group). Sham vs 6 days, $P = 0.0002$; Sham vs 8 days, $P < 0.0001$; Sham vs 14 days, $P < 0.0001$. (E) Protein levels of p300, PPM1K, COL1A, and CTGF in primary proximal tubular epithelial cells (PTECs) treated with TGFβ for indicated time. β-Actin was used as the sample loading control. The graph represents quantification of PPM1K intensity in western blot images, normalized to β-Actin ($n = 3$ per group). Control vs TGFβ(12 h), $P < 0.0001$; Control vs TGFβ(24 h), $P < 0.0001$. (F) Representative images of Masson trichrome staining (MTS) and Sirius red staining of kidney tissue from UUO-induced mouse models infected with Ad5-PPM1K or Ad5-PPM1K N94K (Sham, $n = 5$, UUO, $n = 4$, UUO+Ad5-PPM1K, $n = 5$, UUO+Ad5-PPM1K N94K, $n = 4$). The graph represents quantification of the fibrotic area in histological staining images. Bar $= 100\,\mu m$. MTS: Sham vs UUO, $P < 0.0001$; UUO vs UUO+Ad5-PPM1K, $P = 0.0157$. Sirius red: Sham vs UUO, $P < 0.0001$; UUO vs UUO+Ad5-PPM1K, $P = 0.0107$. (G) mRNA levels of fibrosis-related genes in the kidney tissues from UUO-induced mouse models infected with Ad5-PPM1K or Ad5-PPM1K N94K (Sham, $n = 5$, UUO, $n = 4$, UUO+Ad5-PPM1K, $n = 5$, UUO+Ad5-PPM1K N94K, $n = 4$). *Col1a1*: Sham vs UUO, $P < 0.0001$; UUO vs UUO+Ad5-PPM1K, $P = 0.0003$. *Col3a1*: Sham vs UUO, $P < 0.0001$; UUO vs UUO+Ad5-PPM1K, $P = 0.0001$. *ACTA2*: Sham vs UUO, $P = 0.003$; UUO vs UUO+Ad5-PPM1K, $P = 0.0377$. *FN1*: Sham vs UUO, $P < 0.0001$; UUO vs UUO+Ad5-PPM1K, $P = 0.0018$. Data are presented as mean ± SEM. $*P < 0.05$, $**P < 0.01$, $***P < 0.001$, and $****P < 0.0001$ by ordinary one-way ANOVA. Source data are available online for this figure.

mice but partially recovered in p300 cKO mice. Furthermore, analysis of the endothelial cell integrity of blood vessels showed that the increased albumin leakage in UUO kidneys was significantly reduced in p300 cKO mice compared with WT mice, indicating that loss of p300 in PTCs suppresses UUO-induced EndMT (Fig. 5A). To assess the mesenchymal-transition of endothelial cells in kidney tissues, co-IF

analysis using the endothelial cell marker CD31 and the mesenchymal cell marker vimentin revealed that UUO-induced co-expression of vimentin and CD31 was significantly decreased in p300 cKO mice compared to WT mice (Fig. 5B). Quantitative RT-PCR analysis again verified that EndMT was reduced in UUO p300 cKO mice compared to WT mice (Fig. 5C). To determine the role of PTC-specific p300 in the

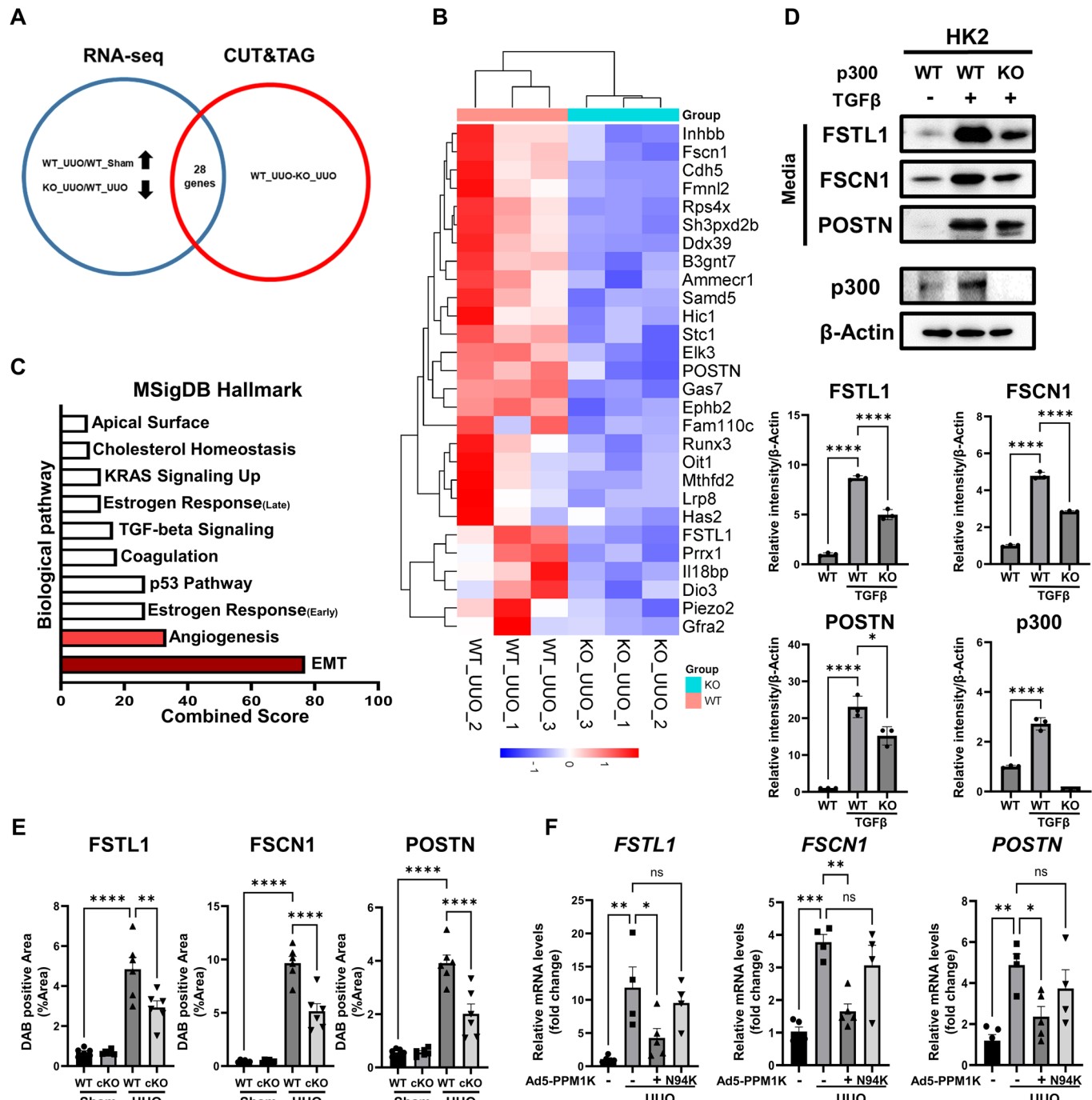

EndMT, we performed a co-culture analysis using HK2 cells and human umbilical vein endothelial cells (HUVECs). TGFβ stimulation of HK2 cells directly induced EndMT in the co-cultured HUVECs (Appendix Fig. S21A–C). Furthermore, conditioned medium (CM) from TGFβ-stimulated HK2 cells induced EndMT in HUVECs. In contrast, CM from p300 knockout HK2 cells resulted in significantly reduced EndMT in HUVECs compared to CM from wild-type (WT) HK2 cells (Appendix Fig. S21D–F). In addition, we analyzed Zonula Occludens-1 (ZO-1), an intercellular tight junction marker, in CD31 positive cells by IF. UUO-induced reduction of intercellular junctions of endothelial

cells was reversed in p300 cKO mouse kidneys compared with WT mouse kidneys (Fig. 5D). Finally, to determine whether the p300-mediated EndMT impaired vascular structure maintenance, we performed three-dimensional imaging of UUO-induced WT and UUO-induced p300 cKO mouse kidneys to examine changes in microvascular structures. We found that the UUO-induced rarefaction of microvascular structures was reversed in p300 cKO mice compared to WT mice (Appendix Fig. S22). Our findings collectively suggest that PTC-specific p300 promotes EndMT, thereby contributing to renal fibrosis.

**Figure 4. PTC-specific p300 mediates transcriptional activation of mesenchymal-transition-related genes to promote renal fibrosis.**

(A) The Venn diagram shows the overlap between differentially expressed genes and direct target genes common to RNA-sequencing and CUT&TAG-sequencing. (B) The 28 genes identified in common by both RNA sequencing and CUT&TAG sequencing. (C) Pathway analysis using MSigDB Hallmark for identify differentially expressed genes in a proximal tubular cell-specific p300 knockout (cKO) mouse model. (D) Protein levels of FSTL1, FSCN1, and POSTN in culture medium from HK2 cells and p300 levels in HK2 cell lysates. Wild-type and p300 knockout (KO) HK2 cells were stimulated by TGFβ. β-actin was used as the sample loading control. The graph represents quantification of intensity in western blot images, normalized to β-Actin ($n = 3$ per group). FSTL1: WT vs WT + TGFβ, $P < 0.0001$; WT + TGFβ vs KO + TGFβ, $P < 0.0001$. FSCN1: WT vs WT + TGFβ, $P < 0.0001$; WT + TGFβ vs KO + TGFβ, $P < 0.0001$. POSTN: WT vs WT + TGFβ, $P < 0.0001$; WT + TGFβ vs KO + TGFβ, $P = 0.0115$. p300: WT vs WT + TGFβ, $P < 0.0001$. (E) The graph represents quantification of FSTL1, FSCN1, and POSTN immunohistochemistry (IHC) images in kidney tissues from wild-type and p300 knockout (cKO) UUO-induced mouse fibrosis model ($n = 6$ per group). FSTL1: WT-Sham vs WT-UUO, $P < 0.0001$; WT-UUO vs cKO-UUO, $P = 0.0041$. FSCN1: WT-Sham vs WT-UUO, $P < 0.0001$; WT-UUO vs cKO-UUO, $P < 0.0001$. POSTN: WT-Sham vs WT-UUO, $P < 0.0001$; WT-UUO vs cKO-UUO, $P < 0.0001$. (F) mRNA levels of FSTL1, FSCN1, and POSTN in kidney tissues from Ad5-PPM1K- or Ad5-PPM1K N94K-infected UUO-induced mouse model kidneys (Sham, $n = 5$, UUO, $n = 4$, UUO+Ad5-PPM1K, $n = 5$, UUO+Ad5-PPM1K N94K, $n = 4$). *FSTL1*: Sham vs UUO, $P = 0.0034$; UUO vs UUO+Ad5-PPM1K, $P = 0.0384$. *FSCN1*: Sham vs UUO, $P = 0.0002$; UUO vs UUO+Ad5-PPM1K, $P = 0.002$. *POSTN*: Sham vs UUO, $P = 0.0022$; UUO vs UUO+Ad5-PPM1K, $P = 0.0344$. Data are presented as mean ± SEM. ns = not significant, *$P < 0.05$, **$P < 0.01$, ***$P < 0.001$, and ****$P < 0.0001$ by ordinary one-way ANOVA. Source data are available online for this figure.

## Selective inhibition of p300 activity suppresses the development of renal fibrosis

We next examined whether inhibition of p300 activity suppressed the development of renal fibrosis by inhibiting the EndMT. To do this, we treated UUO-induced mice with two p300-selective inhibitors, C646 and A6 (Hwang et al, 2020; Kim et al, 2023; Rubio et al, 2023). Both p300 inhibitors significantly suppressed fibrosis progression as determined by MTS, Sirius red staining, and a soluble collagen assay (Figs. 6A and EV4A). Fibrosis-related gene and protein expression were also significantly reduced by both p300 inhibitors (Figs. 6B and EV4B). We confirmed that the effects of these p300 inhibitors were similar in HK2 cells (Fig. EV4C).

Based on our demonstration that p300 promotes the secretion of POSTN, FSTL1, and FSCN1 in renal proximal tubule cells, thereby increasing the EndMT, we next investigated whether selective inhibition of p300 suppressed the secretion of these three proteins in a UUO-induced renal fibrosis mouse model. UUO-induced secretion of the three proteins was significantly reduced by p300 inhibition, especially in the interstitium between PTCs (Fig. 6C; Appendix Fig. S23A). We also confirmed that p300 inhibition suppresses the secretion of the three proteins in HK2 cells (Appendix Fig. S23B). Finally, we observed that the UUO-induced increases in KIM1, albumin leakage, and intercellular junction proteins of endothelial cells were significantly reversed by p300 inhibitor treatment (Fig. 6D; Appendix Fig. S23C). Taken together, these results suggest that selective inhibition of p300 diminishes renal fibrosis by suppressing the EndMT.

## Discussion

CKD, which is accompanied by fibrosis, is a highly prevalent chronic disease with various etiologies that decreases quality of life. The pathogenesis and molecular mechanisms of CKD have therefore been broadly investigated in recent years but are still not clearly understood. Epigenetic dysregulation is known to be a causative factor in several chronic diseases. A few studies on epigenetic dysregulation in kidney disease have been performed, and overexpression of p300 in tubular cells in DN patients has been reported (Gong et al, 2022). However, research on the physiological function of p300 in the development of renal fibrosis remains limited. In this study, we identified increased p300 protein

expression and decreased PPM1K expression in FSGS patients with renal fibrosis compared with MCD patients. Although MCD and FSGS patients both present with nephrotic syndrome, in MCD, glomeruli appear normal under light microscopy whereas some glomeruli in FSGS kidneys show segmental glomerular scarring. Distinction between MCD and FSGS is crucial given that FSGS tends to have a poorer response to treatment and worse prognosis than MCD. Because glomerular scarring in FSGS is focal and segmental, it is possible that cases diagnosed as MCD may actually be unsampled FSGS, and there are no biomarkers to distinguish between MCD and FSGS. Further extensive investigation of p300 expression in CKD patients may enable development of a non-invasive diagnostic method to distinguish MCD from other CKDs.

One of our interesting findings was that when renal fibrosis was induced, the stability of p300 increased without changes in the mRNA expression of p300. We found that AKT mediates phosphorylation of p300 at Ser-1834, thereby increasing the stability of p300 upon induction of renal fibrosis, and conversely, phosphatase PPM1K-mediated de-phosphorylation of p300 leads to de-stabilization of p300. PPM1K is a PPM phosphatase but has not often been reported in association with clinical disease (Gao et al, 2021a; Kamada et al, 2020; Li et al, 2023; Mu et al, 2023). PPM1K was shown to be involved in various physiological pathways such as apoptosis, angiogenesis, cell adhesion, and ECM organization in a human kidney proximal tubule cell line (Zhang et al, 2022). We found that PPM1K dissociated from p300 when renal fibrosis was induced. Conversely, we observed that overexpression of active PPM1K resulted in a reduction in p300 stability and subsequent inhibition of renal fibrosis. In addition, significantly lower expression of PPM1K was identified in FSGS patients than MCD patients. The negative correlation between p300 and PPM1K in CKD highlights the potential of these proteins as biomarkers in advanced CKD patients with fibrosis.

Resident fibroblasts are the major cell type from which ECM-secreting myofibroblasts originate in most organs, but other types of cells also can trans-differentiate into myofibroblasts via the EMT and EndMT. However, recent studies have suggested that the EMT may not play as large a role as expected in the progression of renal fibrosis, and that endothelial cells are the major source of myofibroblasts in renal fibrosis (LeBleu et al, 2013). Furthermore, a recent study reported that the expression of MYC in renal tubular cells promoted fibrosis by inducing the EndMT (Lovisa et al, 2020). In accordance with the above reports, we demonstrated that the

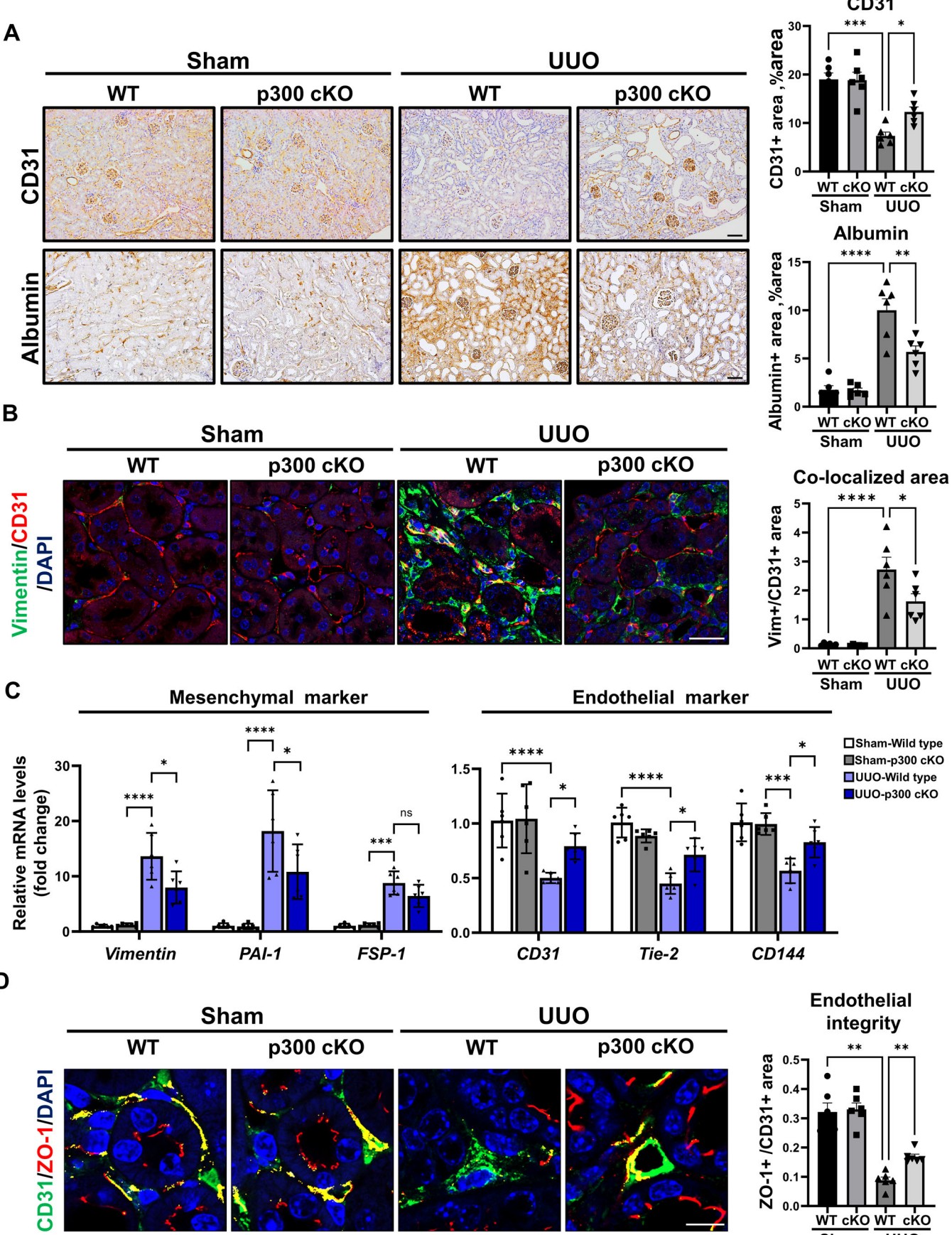

**Figure 5. PTC-specific p300 mediates the EndMT via secretion of mesenchymal-transition-related proteins in renal fibrosis.**

(A) Representative images of CD31 and albumin immunohistochemistry (IHC) in kidney tissues from wild-type and p300 knockout (cKO) UUO-induced mouse fibrosis model. The graph represents quantification of the DAB-positive area in histological staining images ($n = 6$ per group). Bar = 100 μm. CD31: WT-Sham vs WT-UUO, $P = 0.0003$; WT-UUO vs cKO-UUO, $P = 0.0187$. Albumin: WT-Sham vs WT-UUO, $P < 0.0001$; WT-UUO vs cKO-UUO, $P = 0.0018$. (B) Representative images of CD31 and vimentin co-localization (IF) in kidney tissues from wild-type and p300 knockout (cKO) UUO-induced mouse fibrosis model. The graph represents quantification of the co-localized area ($n = 6$ per group). Bar = 50 μm. WT-Sham vs WT-UUO, $P < 0.0001$; WT-UUO vs cKO-UUO, $P = 0.0285$. (C) mRNA levels of endothelial and mesenchymal cell marker genes in kidney tissues from wild-type and p300 knockout (cKO) UUO-induced mouse fibrosis model ($n = 6$ per group). *Vimentin*: WT-Sham vs WT-UUO, $P < 0.0001$; WT-UUO vs cKO-UUO, $P = 0.0127$. *PAI-1*: WT-Sham vs WT-UUO, $P < 0.0001$; WT-UUO vs cKO-UUO, $P = 0.0007$. *FSP-1*: WT-Sham vs WT-UUO, $P = 0.0003$. *CD31*: WT-Sham vs WT-UUO, $P < 0.0001$; WT-UUO vs cKO-UUO, $P = 0.0129$. *Tie-2*: WT-Sham vs WT-UUO, $P < 0.0001$; WT-UUO vs cKO-UUO, $P = 0.0282$. *CD144*: WT-Sham vs WT-UUO, $P < 0.0001$; WT-UUO vs cKO-UUO, $P = 0.0296$. (D) Representative image of CD31 and ZO-1 co-immunofluorescence (IF) in kidney tissue from wild-type and p300 knockout (cKO) UUO-induced mouse fibrosis model. The graph represents quantification of the co-localized area of ZO-1 and CD31 in co-immunofluorescence (IF) images of kidney tissues from wild-type and p300 knockout (cKO) UUO-induced mouse fibrosis model ($n = 6$ per group). Bar = 25 μm. WT-Sham vs WT-UUO, $P = 0.002$; WT-UUO vs cKO-UUO, $P = 0.0018$. Data are presented as mean ± SEM. $*P < 0.05$, $**P < 0.01$, $***P < 0.001$, and $****P < 0.0001$ by ordinary one-way ANOVA. Source data are available online for this figure.

progression of EndMT, as well as EMT, in renal fibrosis was mediated by PTC-specific p300 protein. Moreover, we unraveled the involvement of mesenchymal-transition-related secretory proteins whose secretion was regulated by PTC-specific p300 using in vitro and in vivo experiments. We also observed that EndMT induced microvascular rarefaction, suggesting that the resulting deficiency in tissue oxygen and nutrient supply could be alleviated by p300 inhibition. However, the precise functions of the secreted proteins and the mechanisms by which signals are transmitted to endothelial cells, leading to mesenchymal transition, remain unclear. Furthermore, our understanding of how the combined actions of EMT and EndMT contribute to the progression of fibrosis is insufficient. Therefore, detailed follow-up studies are required to elucidate these mechanisms. In addition, endothelial-specific p300 has been reported to play an important role in mediating liver fibrosis, and our results also showed a slight increase in endothelial p300 levels, though less than in proximal tubule cells, suggesting that additional studies are needed to elucidate the physiological role and molecular mechanisms of endothelial p300 in renal fibrosis (Gao et al, 2021b).

Notably, several clinical trials of anti-fibrotic molecules such as FG-3019, pirfenidone, and bardoxolone methyl have been conducted in CKD patients (Yamashita and Kramann, 2024). However, these compounds target a single molecule/cell type and have no noticeable therapeutic effect. In actuality, fibrosis is caused by the cooperative action of many cell types such as epithelial cells, immune cells, and fibroblasts, thus, the development of treatments targeting single molecules/cell types is likely to encounter limitations. There are several lines of evidence that p300 is a highly promising fibrosis treatment target. First, p300 regulates multiple pro-fibrotic genes related to the EndMT, angiogenesis, and inflammation, and p300 may therefore be a plausible target for treating fibrotic disease in CKD patients (Ruiz-Ortega et al, 2022). Second, elevated p300 expression has been commonly observed in fibrotic regions of various organs, including the lungs and kidneys (Gong et al, 2022; Lee et al, 2023). Moreover, the development of renal fibrosis in the mouse model was associated with a significant increase in p300 protein expression among other HAT proteins. Therefore, therapies targeting p300 might be more specifically targeted to the fibrotic region and exhibit high anti-fibrotic efficacy.

In this study, we verified the anti-fibrotic effects of p300 inhibitors through the suppression of EndMT in a renal fibrosis model. However, at this stage, we cannot exclude the possibility that p300 inhibition regulates fibrosis not only through the suppression of EndMT via the inhibition of transcriptional activation of mesenchymal-transition-related secretory proteins but also through the direct suppression of EMT. Additionally, mesenchymal-transition-related secretory proteins may influence tubular epithelial cells as well. Thus, further preclinical studies are needed to assess the therapeutic efficacy of selective p300 inhibitors and to elucidate their pharmacological mechanisms in the treatment of CKD with fibrosis.

We here demonstrated that the p300 protein in PTCs mediates renal fibrosis by promoting the secretion of target proteins, which consequently leads to induction of the EndMT of endothelial cells. In particular, we demonstrated that dissociation of PPM1K from p300 increases p300 stability, which leads to EndMT and renal fibrosis induction. Together, our findings provide the basis for the development of CKD therapy by selective inhibition of p300.

## Methods

### Reagents and tools table

| Reagent/Resource | Reference or Source | Identifier or Catalog Number |
|---|---|---|
| **Experimental models** | | |
| HK2 cell line | ATCC | CRL-219 |
| HUVEC | ATCC | CRL-1730 |
| Mouse primary PTC | Male C57BL/6 kidney | NA |
| C57BL/6 | Orient Bio | NA |
| γGT-1 Cre mice | Jackson Laboratory | #012841,RRID:IMSR_JAX:012841 |
| p300 flox mice | Jackson Laboratory | #:025168, RRID:IMSR_JAX:025168 |
| **Recombinant DNA** | | |
| Phosphatase-plasmid | Park et al, 2017 | NA |
| PSG5-p300-HA-plasmid | Addgene | 89094, RRID:Addgene_89094 |
| PX549 for CRSPR-cas9-Plasmid | Addgene | 62988, RRID:Addgene_62988 |

| Reagent/Resource | Reference or Source | Identifier or Catalog Number |
|---|---|---|
| Adeno-X- ZsGreen1 | TAKARA | 632269 |
| **Antibodies** | | |
| Antibody | This Study | Appendix Table S4 |
| **Oligonucleotides and other sequence-based reagents** | | |
| Mutagenesis primer | This Study | Appendix Table S3 |
| PCR primer | This Study | Appendix Table S5 |
| **Chemicals, Enzymes and other reagents** | | |
| TGF-beta1 | ProSpec | CYT-716 |
| C646 | Selleckchem | S7152 |
| MG132 | Sigma-Aldrich | M7449 |
| Cycloheximide | Sigma-Aldrich | C4859 |
| A6 | Hwang et al, 2020 | NA |
| Sircol collagen assay | Biocolor Ltd | S1000 |
| Proximity ligation assay | Sigma-Aldrich | DUO92101 |
| RT-qPCR SYBR mixture | Roche | 04913914001 |
| Ad5-Adenovrius for mammalian overexpression | This Study | NA |
| SCARF-clear | Crayon tech | 20-3021 |
| C-MATCH | Crayon tech | 50-3010 |
| TNT T7 quick coupled transcription/ translation kit | Promega | L1170 |
| RNAiso Plus | TAKARA | 9109 |
| Zenon Labeling reagent | Thermo Fisher Scientific | Z25302 |
| Envision+ system-HRP Labeled Polymer anti-rabbit | Dako | K4003 |
| Antigen Unmasking solution | Vector | H-3300-250 |
| 3'-diaminobenzidine substrate | Vector | SK-4100 |
| Picro-Sirius Red | Abcam | ab150681 |
| MTS | Sigma-Aldrich | HT15 |
| Wiegert's Iron Hematoxylin | Sigma-Aldrich | HT1079 |
| Folic acid | Sigma-Aldrich | 172502 |
| Lipofectamine 3000 | Thermo Fisher Scientific | L3000015 |
| RNAiMAX | Thermo Fisher Scientific | 13778075 |
| **Software** | | |
| Prism 10 | Graphpad | NA |
| GREAT | https://great.stanford.edu/great/public/html/ | NA |

| Reagent/Resource | Reference or Source | Identifier or Catalog Number |
|---|---|---|
| Intergrative Genomics View | https://igv.org/ | NA |
| Cellenic® | Biomage | NA |
| CELLXGENE discover | CellxGene | NA |
| **Other** | | |
| CKD patients kidney sample | Severance Hospital tissue bank | NA |
| Normal human kidney samples | TissueArray.Com LLC | KD1002 |

## Clinical sample of chronic kidney disease (CKD) patients

Kidney tissue sample of CKD were obtained from the Severance Hospital tissue bank and this research was approved by the Ethics Committee of the Institutional Review Board (IRB) of Severance Hospital (protocol no. 4.2024.0314). Informed consent was obtained from all human participants, and the study was conducted in accordance with the principles of the WMA Declaration of Helsinki and the Department of Health and Human Services Belmont Report. Kidney tissues from CKD patients with Minimal Change Disease (MCD), Focal segmental glomerulosclerosis (FSGS) and normal kidney tissue purchased from TissueArray.Com LLC were included in this study. Detailed information of CKD patients such as, age, sex, diagnosis, lab results, is included in Appendix Table S1–2. The kidney tissue samples were collected from patients who had biopsies performed at Severance Hospital (Seoul, Korea) between March 2005 and December 2022 and among the cases diagnosed with FSGS, those with a potential for secondary FSGS were strictly excluded to secure as many samples as possible. CKD patients were diagnosed by institutional diagnostic criteria including blood BUN, urine protein, creatinine, histopathologic classification, Stanford classification, MEST-C scoring, etc and each case was grouped based on clinical diagnosis (excluding other variables such as gender and age). To ensure patient privacy and eliminate bias during the study, names were coded.

## Cell culture

Human proximal tubule cell line (HK2) and human umbilical vein endothelial cell (HUVEC) were purchased from American Type Culture Collection (Manassas, VA, USA). HK2 were maintained in RPMI1640 (Corning Inc., Corning, NY, USA) supplemented with 10% FBS and 1% antibiotic-antimitotic solution. HUVEC were maintained in Endothelial Cell medium (Sciencell Research Laboratories, Carlsbad, CA, USA). Primary proximal tubular cells (PTEC) were isolated from mouse kidney and maintained in RPMI1640 (Corning, NY, USA) supplemented with 10% FBS and 1% antibiotic-antimitotic solution. All cells were incubated at 37 °C and 5% $CO_2$. HK2 and PTEC were starved in no-serum media for 2 h and then stimulated with 20 ng/ml TGFβ1 (Prospec, Rehovot, Israel). Concentration of cell

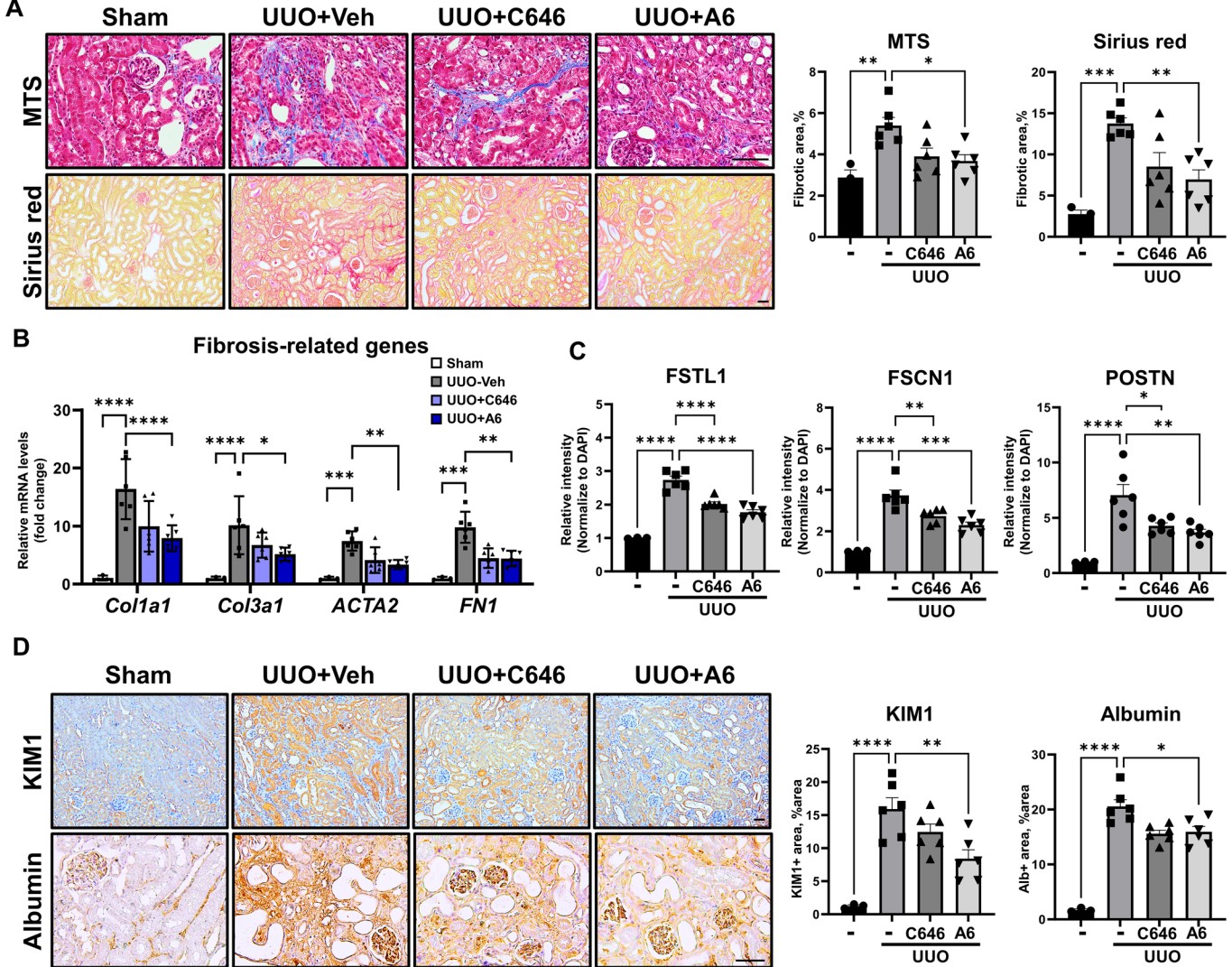

**Figure 6. Selective inhibition of p300 suppresses renal fibrosis.**

(A) Representative image of Masson trichrome staining (MTS) and Sirius red staining of kidney samples from UUO-induced fibrosis mice injected with C646 and A6. The graph represents quantification of the fibrotic area (sham, $n = 3$, UUO+Veh, $n = 6$, UUO + C646, $n = 6$, UUO + A6, $n = 6$). Bar = 100 μm. MTS: Sham vs UUO, $P = 0.0039$; UUO vs UUO + A6, $P = 0.0161$. Sirius red: Sham vs UUO, $P = 0.0003$; UUO vs UUO + A6, $P = 0.0041$. (B) mRNA levels of fibrosis-related genes in kidney tissues from the UUO-induced fibrosis mouse model injected with C646 and A6 (sham, $n = 3$, UUO+Veh, $n = 6$, UUO + C646, $n = 6$, UUO + A6, $n = 6$). *Col1a1*: Sham vs UUO, $P < 0.0001$; UUO vs UUO + A6, $P < 0.0001$. *Col3a1*: Sham vs UUO, $P < 0.0001$; UUO vs UUO + A6, $P = 0.0121$. *ACTA2*: Sham vs UUO, $P = 0.0001$; UUO vs UUO + A6, $P = 0.0016$. *FN1*: Sham vs UUO, $P = 0.0001$; UUO vs UUO + A6, $P = 0.0057$. (C) The graph represents quantification of the intensity of FSTL1, FSCN1, and POSTN in co-immunofluorescence (IF) images of kidney tissues from the UUO-induced fibrosis mouse model injected with C646 and A6, normalized to DAPI (sham, $n = 3$, UUO+Veh, $n = 6$, UUO + C646, $n = 6$, UUO + A6, $n = 6$). Bar = 100 μm. FSTL1: Sham vs UUO, $P < 0.0001$; UUO vs UUO + C646, $P < 0.0001$; UUO vs UUO + A6, $P < 0.0001$. FSCN1: Sham vs UUO, $P < 0.0001$; UUO vs UUO + C646, $P = 0.0064$; UUO vs UUO + A6, $P = 0.0002$. POSTN: Sham vs UUO, $P < 0.0001$; UUO vs UUO + C646, $P = 0.0143$; UUO vs UUO + A6, $P = 0.003$. (D) Representative image of KIM1 and Albumin immunohistochemical staining of kidney tissues from the UUO-induced fibrosis mouse model injected with C646 and A6. The graph represents quantification of the DAB positive area (sham, $n = 3$, UUO+Veh, $n = 6$, UUO + C646, $n = 6$, UUO + A6, $n = 6$). Bar = 100 μm. KIM1: Sham vs UUO, $P < 0.0001$; UUO vs UUO + A6, $P = 0.0053$. Albumin: Sham vs UUO, $P < 0.0001$; UUO vs UUO + A6, $P = 0.0124$. Data are presented as mean ± SEM. *$P < 0.05$, **$P < 0.01$, ***$P < 0.001$, and ****$P < 0.0001$ by ordinary one-way ANOVA. Source data are available online for this figure.

cultured medium were performed using VIVASPIN 20 (SATOROUS, Göttingen, Germany) following the manufacturer's protocol. Transient transfection was performed using Lipofectamine 3000 and RNAiMAX (Thermo Fisher Scientific, Waltham, MA, USA). HK2 and HUVEC cell co-culture systems were organized using transwell (Corning Inc., Corning, NY, USA). HK2 were treated by 3 μM C646 (Selleckchem LLC, TX, USA) or 3 μM A6 and then collected 24 h after treatment. The plasmid we used for p300 overexpression was cloned

into PSG5-HA and the plasmid used for phosphatase overexpression was cloned into PSG5-Flag. The siRNA of PPM1K used to knock-down PPM1K was produced by Genolution (Seoul, Korea). Sequence of siPPM1K #1:5′-GGGAUAACCGCAUUGAUGAUU-3′, 5′-UCAU CAAUGCGGUUAUCCCUU-3′ siPPM1K #3:5′-CUUCCUAAGGA GAAGAACUUU-3′, 5′-AGUUCUUCUCCUUAGGAAGUU-3′. All cell lines were authenticated by short tandem repeat (STR) profiling and confirmed negative for mycoplasma contamination.

## Animal model

All animal experiments were approved by Yonsei University College of Medicine Institutional Animal Care and Use Committee (IACUC No. 2023-0025), and were conducted following the ARRIVE guidelines. Eight-week-old C57BL/6 mice were purchased from ORIENT BIO Inc. (Seongnam, Korea). Purchased mice underwent a 2-week acclimatization period. The mice were maintained in specific pathogen free condition (SPF) and 12-h light/dark cycle conditions. Mice with conventionally p300-deficient from kidney proximal tubular cells, were generated by intercrossing γGT-1 Cre mice and p300 Floxed mice. These mice were purchased from The Jackson Laboratory (Bar Harbor, ME, USA). Unilateral Ureteral Obstruction (UUO) surgery was performed under anesthesia with incision in the lateral abdomen and ligation of a single ureter. Folic acid (FA) administration was performed by FA dissolved in Sodium-Bicarbonate, which was injected intra-peritoneally once at 250 mg/kg. The streptozotocin-unilateral nephrectomy (STZ-UNx) model was established by first performing open unilateral nephrectomy surgery after anesthesia, and 1 week later, 50 mg/kg STZ was injected intra-peritoneally for 5 days. STZ was dissolved in 0.025 M sodium citrate and the mice were validated by checking fasting blood glucose (>300 mg/dL). The Adriamycin (ADR) nephropathy model was induced by a single injection of ADR at a dose of 20 mg/kg via the tail vein, and mice were sacrificed 6 weeks after injection. PPM1K overexpression mice via adenovirus infection were introduced with Ad5-GFP-Flag-PPM1K ($1 \times 10^{10}$ vp) by direct sub-capsular injection into the kidney, and viral infection was confirmed by detecting GFP in the kidney tissue. Anesthesia was induced by Zoletil (50 mg/ml) and Rumpun (23.32 mg/ml). Both male and female mice were randomly assigned and used in the experiments. All mice were randomly selected for experiments within the same age group. Also, all animal experiments were conducted in a blinded manner to minimize bias. The non-experimental group underwent sham surgery and vehicle treatment.

## CRSPR-Cas9

The p300-deficient HK2 cells were engineered via the Crispr-cas9 method. The sgRNA (5′-GTACGACTAGGTACAGGCGA-3′) designed by Chop-Chop was inserted into the px549 plasmid vector. Vectors were introduced into HK2 cells via electroporation and selected by puromycin treatment. And the expression of p300 was confirmed by RT-qPCR and western blot.

## Site direct mutation

The mutant constructs of p300 and PPM1K were created by DNA polymerases master mixes (Thermo Fisher Scientific, Waltham, MA, USA). All constructs were verified by DNA sequencing from Macrogen Co. (Seoul, Korea). Used primer information is listed in Appendix Table S3.

## Histological staining

Histologic staining was performed using de-paraffinized kidney tissue. Masson trichrome staining was performed via Wiegert's Iron Hematoxylin kit (Sigma-Aldrich, Burlington, MA, USA) and

MTS Kit (Sigma-Aldrich, Burlington, MA, USA) following the manufacturer's protocol. Picro-sirius red staining was performed via the Picro-Sirius Red Stain Kit (Abcam, Cambridge, UK). Kidney sections were submerged in Picro-Sirius Red solution for 2 h at room temperature, washed twice with 5% acidified water, and then washed twice with absolute alcohol. After staining, sections were mounted using a hydrophilic mounting solution. To perform the Immunohistochemistry (IHC) and immunofluorescence staining (IF). The de-paraffinized kidney sections has been treated with antigenic revitalization by Antigen Unmasking solution (Vector Laboratories, Burlingame, CA, USA) in the pressure cooker at 5 min. Peroxidase blocking step was performed by Dako REAL Peroxidase-blocking Solution (Dako, Glostrup, Denmark). Unmasked and peroxidase blocked kidney tissue were incubated with a primary antibody against p300, PPM1K, CD31, AQP1, AQP2, WT1, FSTL1, FSCN1, POSTN, Albumin, KIM-1, Vimentin, Col1A, all of which were used at a dilution of 1:200. For IHC, primary antibody staining was followed by HRP-conjugated anti-rabbit secondary antibody using Envision+ system-HRP Labeled Polymer anti-rabbit (Dako, Glostrup, Denmark) and visualized by 3′- diaminobenzidine substrate (Vector Laboratories, CA, USA). For IF, Dylight 488 anti-mouse antibody (1:500, Vector Laboratories, CA, USA) and Dylight 549 anti-rabbit anti-body (1:500, Vector Laboratories, CA, USA) were used for fluorescent labeling. To immunostain with antibodies whose host is mouse, they were initially fluorescently labeled with Zenon kit. To immune-stain with anti-mouse antibody, the antibody was initially fluorescently labeled with Zenon Labeling Kits following the manufacturer's protocol. Detailed antibody information is listed in Appendix Table S4. Quantification of IHC were assessed by ImageJ and co-localization were assessed by ZEN application.

## RNA isolation and reverse transcription-quantitative polymerase chain reaction

Total RNA was extracted from kidney tissues and cells using the RNAiso Plus reagent (TaKaRa Bio, Otsu, Japan) following the manufacturer's instructions. The RNA quantity and purity were analyzed by a NanoDrop 2000 spectrophotometer (Thermo Fisher Scientific, MA, USA). Extracted RNA was reverse-transcribed using Cell-script (CellSafe, Yongin, Korea) following the manufacturer's instructions. qRT-PCR analyses were performed using FastStart Universal SYBR Green Master (ROX) reagents (Roche, Basel, Switzerland) and ABI Prism 7700 (Applied Biosystems, CA, USA). Target gene expression levels were normalized by comparison to GAPDH expression levels. All reactions were performed in triplicate. The detailed primer information used for amplification is listed in Appendix Table S5.

## Western blot

Kidney tissue and cells were lysed by Lysis Buffer (Triton X-100 0.2%, EGTA 1 mM, NaF 1 mM, NaCl 150 mM, $Na_3VO_4$ 1 mM, EDTA 1 mM, 50 mM Tris HCl pH 7.4, Xpert proteinase inhibitor cocktail (Gene depot, TX, USA)). Protein lysates were separated by centrifugation at 12,000 rpm at 4 °C for 15 min and then boiled with 5X SDS sample buffer (20% SDS, 50% Glycerol, 0.1% bromophenol blue solution, 0.5% 2-mercaptoethanol) at 95 °C for 5 min. Boiled protein samples were separated by electrophoresis on

a polyacrylamide gel and gel was transferred to NC (GE Healthcare, IL, USA) membranes. The following primary antibodies were used: Anti-p300, Anti-PPM1K, Anti-AQP-1, Anti-AKT, Anti-p-AKT, Anti-p-p300, Anti-Col1a, Anti-CTGF, Anti-FSTL1, Anti-FSCN1, Anti-POSTN, Anti-alpha-SMA, Anti-p53, Anti-HA, Anti-Flag and Anti-beta-actin. All primary antibodies were used at a dilution of 1:1000. The secondary antibodies used were: anti-mouse HRP-antibody (1:5000, 31430, Thermo Fisher Scientific, MA, USA) and anti-rabbit HRP-antibody (1:5000, 31460, Thermo Fisher Scientific, MA, USA). Chemiluminescence signals were visualized on the Fusion SOLO S device (Vilber, Marne-la-Vallée, France) and quantified using FIJI-Image-J software. Detailed antibody information is listed in Appendix Table S5.

## Immunoprecipitation assay

Cells were lysed by Lysis BF (Triton X-100 0.2%, EGTA 1 mM, NaF 1 mM, NaCl 150 mM, $Na_3VO_4$ 1 mM, EDTA 1 mM, 50 mM Tris HCl pH 7.4, Xpert proteinase inhibitor cocktail (Gene Depot, TX, USA)). Protein lysates were separated by centrifugation at 12,000 rpm at 4 °C for 15 min. A 10% input sample was remained and the lysates were rotated with HA-magnetic beads (Sigma-Aldrich, MA, USA) for 24 h at 4 °C. After rotation, the lysate and bead mixture was placed on a magnetic rack and the supernatant was removed. The beads were washed with lysis BF for 20 min RT and then placed back on the magnetic rack. Repeat the previous washing process 3 times. Proteins were eluted in 1x sodium dodecyl sulfate buffer (20% SDS, 50% Glycerol, 0.1% bromophenol blue solution, 0.5% 2-mercaptoethanol) after boiling for 10 min at 95 °C.

## In vitro translation system

In vitro transcription and translation were performed using a TNT T7 quick coupled transcription/translation kit following the manufacturer's protocol (Promega Corporation, Madison, WI, USA).

## Kidney functional test

Blood samples were collected from the heart of anesthetized mice, and the blood was separated to the serum by centrifugation. BUN (Blood Urea Nitrogen) and serum Creatinine were measured with a biochemistry analyzer (Dri-chem NX500, FUJIFILM, Tokyo, Japan).

## Proximity ligation assay (PLA)

PLA was performed using the Duolink® Proximity Ligation Assay kit. The assay was performed according to the manufacturer's protocol and used Anti-p300 (1:100, SC48343, Santa Cruz Biotechnology, Dallas, TX, USA), Anti-PPM1K antibodies (1:100, ab135286, Abcam, Cambridge, UK). Cells were prepared by fixation in 4% paraformaldehyde and permeabilizing with Triton X-100. Paraffin-embedded tissues were de-paraffinized with xylene and hydration with 100%, 90%, 80% EtOH, before permeabilization with Triton X-100. The samples were mounted with Duo-link mounting Sol. and observed by Zeiss LSM 900 microscope.

## 3-Dimensional imaging

For 3-Dimensional imaging, a Paraformaldehyde-fixed 1 $mm^3$ section of renal cortex was cleared using SCARF-clear buffer and C-MATCH buffer on a C-Stain device (Crayon Tech., Gyeong-gi-Do, Korea). Cleared kidney tissue was immersed in a solution of CD31 (ab28364, Abcam, Cambridge, UK) antibody diluted 1:250 in 3% BSA with sodium azide and shaking at 37 °C for 3 days. After washing with BSA three times, kidney tissue was immersed in a solution of DyLight-649 Anti-rabbit antibody diluted 1:200 in 3% BSA with sodium azide and shaking at 37 °C for 2 days. Stained tissue was mounted using C-MATCH solution in a confocal dish. Images and videos were acquired using a Zeiss LSM 900 instrument (Carl Zeiss, Baden-Württemberg, Germany), and image processing and quantification were performed using Zeiss' ZEN blue program.

## Sircol collagen assay

Fresh kidney with no-capsule digested by 0.1 mg/ml pepsin in acetic acid at room temperature overnight and then centrifuged at 12,000 rpm for 20 min. The content of collagen in kidney is measured according to the instructions for a Sircol collagen assay kit (Biocolor, Carrickfergus, UK). Briefly, 50 ml of tissue-free supernatant of pepsin-digested kidney and the standards are incubated with 1 ml of Sircol dye reagent for 30 min on a rotary table. After centrifugation and washing with 750 ml of ice-cold acid-salt wash reagent, the collagen-dye pellet is suspended with 500 ml of alkali reagent and then samples are read at 555 nm by adding 200 ml of the collagen supernatant in alkali reagent into 96-well plates. The content of collagen in each sample is calculated according to the standard curve.

## Public data analysis

To determine gene expression levels in the Gene expression omnibus (GEO) dataset (GSE142025, GSE212681) (Fan et al, 2019; Zhang et al, 2022), we used GEO2R web-based tool provided by NCBI. The ChIP-sequencing (GSE75948) data and ATAC-sequencing (GSE197815) data were then visualized using the Intergrative Genomics View (https://igv.org/) application to identify promoter enrichment and chromatin accessibility (Cao et al, 2022; O'Brien et al, 2016). Single-cell RNA-sequencing of CKD patients (GSE183279) were analyzed by Cellenic® provide by Biomage (https://www.biomage.net/) and UMAPs of single-cell RNA-sequencing were visualized using CELLXGENE discover (https://cellxgene.cziscience.com/).

## RNA-sequencing

For bulk RNA-sequencing, RNA extracted with the QIAGEN RNeasy® kit from cortex sections of kidneys was used. RNA purity was measured with a Nanodrop2000 (Thermo Fisher Scientific, MA, USA). RNA-sequencing was performed and analyzed on Macrogen Co. (Seoul, Korea) with QuantSeq 3' mRNA-Seq protocol (UCSC, mm10). Pathway analysis was performed using PANTHER-based GO enrichment analysis (MSigDB Hallmark) from the Gene Ontology (GO) Resource (https://geneontology.org/)98. The heatmap and pathway dot graph were visualized using SRplot- Science and Research online plot (bioinformatics.com.cn).

**The paper explained**

**Problem**

Kidney fibrosis is a chronic disease caused by various factors, characterized by the deposition of extracellular matrix components such as collagen. However, the underlying mechanisms of kidney fibrosis remain unclear. Identifying common pathogenic mechanisms of kidney fibrosis could provide crucial insights for developing effective therapeutic strategies.

**Results**

We observed that histone acetyltransferase p300 is elevated in proximal tubular cells of CKD patient kidneys and mouse models of kidney fibrosis. We also found that p300 in proximal tubular cell is regulated by the phosphatase PPM1K. Furthermore, p300 in proximal tubular cells promotes fibrosis by inducing endothelial-to-mesenchymal transition. Notably, treatment with p300-specific inhibitors demonstrated antifibrotic effects in a mouse model of kidney fibrosis.

**Impact**

Our results demonstrate the role of p300 in the development of renal fibrosis, and suggest that p300 is a promising target for treatment of advanced CKD.

## CUT&TAG analysis

To perform CUT&TAG analysis, mouse kidney proximal tubular cells were initially isolated. Using the gentleMACS™ Dissociator (Miltenyi Biotec, Bergisch Gladbach, Germany). The kidneys were minced by Brain_03, Spleen_04 programs, and RBCs were removed using Red Blood Cell Lysis Solution (Miltenyi Biotec, NRW, Germany) and minced tissue were incubated with 1:500 Fluorescein-Tetragonolobus Lectin (FL-1321-2, Vector laboratories, CA, USA) for 2 h at room temperature. BD FACS Aria III was used for cell isolation by Flow cytometry. FITC positive cells were isolated using BD FACS Aria III and confirmed as PTCs by RT-qPCR and IF. Using the isolated PTCs, we generated sequencing-ready libraries by CUT&TAG analysis kit (Active Motif, CA, USA) according to the manufacturer's protocol using an anti-rabbit p300 antibody (CS86377, Cell Signaling Technology, MA, USA). CUT&TAG products were sequenced and analyzed on a HiSeq2500 instrument (Illumina Inc., CA, USA) at Macrogen Co. (Seoul, Korea) and Gene annotation analysis was performed using the GREAT web-based tool (https://great.stanford.edu/great/public/html/, UCSC mm10).

## Statistical analysis

Statistical analyses were performed by one-way ANOVA (comparisons of more than two groups, using Mann Whitney or Kruskal Wallis analyses) and T-test analyses (comparisons of two groups). Correlation analysis was calculated by simple linear regression analyses. We performed a Shapiro-Wilk test for datasets with $n \leq 30$, confirming normality. For datasets with $n > 30$, normality testing was omitted based on the Central Limit Theorem (CLT). All data were included in the analysis, and no exclusion criteria were applied. Statistical analyses were performed using GraphPad Prism 10 (GraphPad Software Inc, La Jolla, CA, USA). A $p$-value of <0.05 was considered a statistically significant difference.

## Data availability

The RNA-sequencing data have been deposited in the Gene Expression Omnibus (GEO) (GSE271305) and The Cut and Tag-analysis data have been deposited in the NCBI Sequence Read Archive (SRA) (PRJNA1130096). GSE271305 data can be found at GEO using the following link: https://www.ncbi.nlm.nih.gov/geo/query/acc.cgi?acc=GSE271305. PRJNA1130096 data can be found at SRA using the following link: https://www.ncbi.nlm.nih.gov/bioproject/1130096.

The source data of this paper are collected in the following database record: biostudies:S-SCDT-10_1038-S44321-025-00243-1.

## Peer review information

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

## Acknowledgements

This research was supported by the National Research Foundation of Korea (NRF) grant funded by the Korean government (MSIT) (No. RS-2025-02214844 to H.G.Y. and No. NRF-2022R1A2C1091055 to J.Y.Y.), and by a research grant from Gangnam Severance Hospital, Yonsei University College of Medicine.

## Author contributions

**Hyunsik Kim**:Conceptualization; Data curation; Formal analysis; Investigation; Visualization; Methodology; Writing—original draft. **Soo-Yeon Park**:Data curation; Supervision; Investigation; Methodology. **Soo Yeon Lee**:Methodology. **Jae-Hwan Kwon**:Investigation; Methodology. **Seunghee Byun**:Methodology. **Byounghwi Ko**:Resources. **Jung-Yoon Yoo**:Investigation; Methodology. **Beom Seok Kim**:Supervision; Project administration; Writing—review and editing. **Beom Jin Lim**:Supervision; Project administration; Writing—review and editing. **Ho-Geun Yoon**:Conceptualization; Data curation; Supervision; Project administration.

Source data underlying figure panels in this paper may have individual authorship assigned. Where available, figure panel/source data authorship is listed in the following database record:biostudies:S-SCDT-10_1038-S44321-025-00243-1.

## Disclosure and competing interests statement

The authors declare no competing interests.ll

# Expanded View Figures

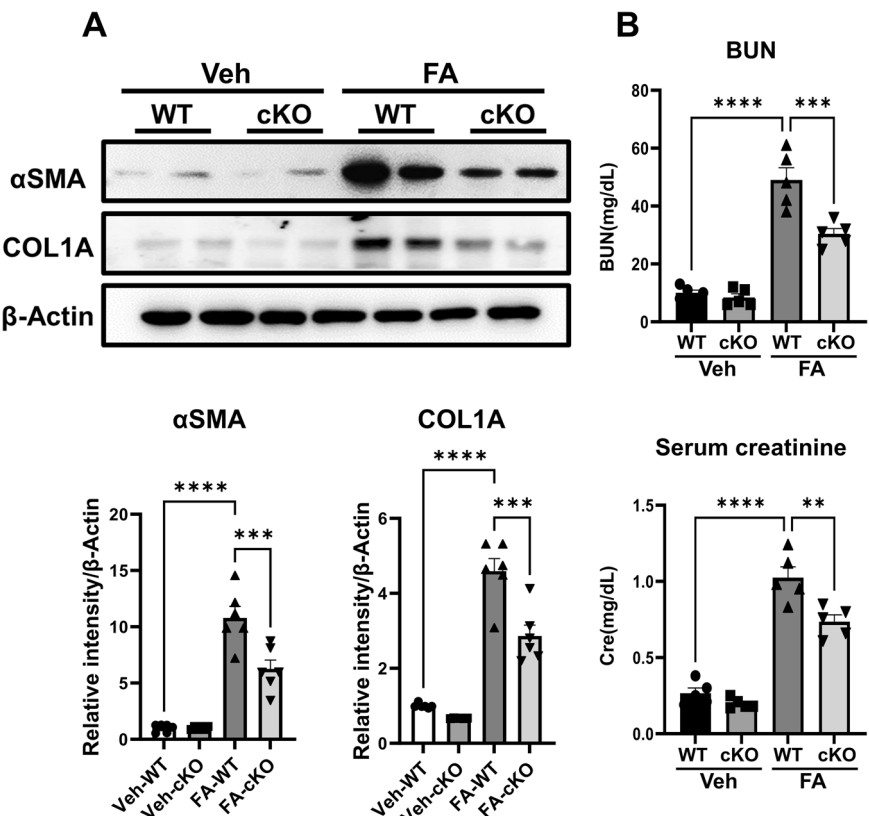

**Figure EV1. Evaluation of fibrosis-related proteins and renal function in p300 cKO FA-induced fibrosis mouse models.**

(**A**) Protein levels of fibrosis markers, αSMA and COL1A, in kidney tissues from wild-type and p300 knock-out (cKO) FA-induced fibrosis mouse models. β-Actin was used as the loading control. The graph represents the quantification of intensity in western blot images, normalized to β-Actin ($n = 6$ per group). αSMA: Veh-WT vs FA-WT, $P < 0.0001$; FA-WT vs FA-cKO, $P = 0.0004$. COL1A: Veh-WT vs FA-WT, $P < 0.0001$; FA-WT vs FA-cKO, $P = 0.0001$. (**B**) Renal function was assessed using serum samples from wild-type and p300 knock-out (cKO) FA-induced fibrosis mouse models ($n = 5$ per group). BUN: Veh-WT vs FA-WT, $P < 0.0001$; FA-WT vs FA-cKO, $P = 0.0003$. Serum creatinine: Veh-WT vs FA-WT, $P < 0.0001$; FA-WT vs FA-cKO, $P = 0.0021$. Data are presented as mean ± SEM, $**P < 0.01$, $***P < 0.001$, and $****P < 0.0001$ by ordinary one-way ANOVA. Source data are available online for this figure.

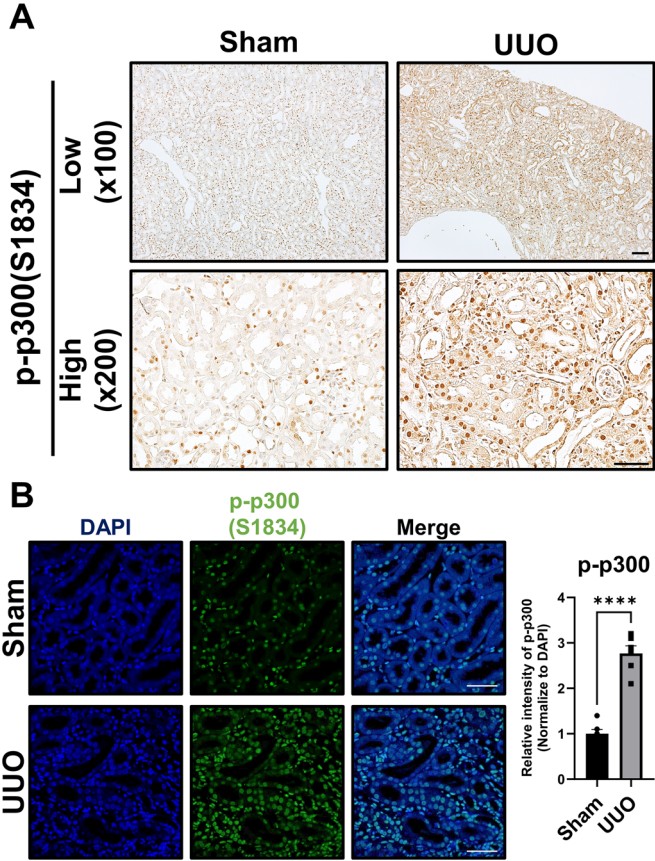

**Figure EV2. Increase of p300 phosphorylation at Serine 1834 in kidney tissue from the UUO-induced mouse fibrosis model.**

(A) Representative image of phosphorylated p300 immunohistochemistry (IHC) in mouse kidney tissue from the UUO-induced mouse fibrosis model. Bar = 100 μm. (B) Representative image of phosphorylated p300 immuno-fluorescence (IF) in mouse kidney tissue from the UUO-induced mouse fibrosis model. The graph represents the quantification of intensity of p-p300 S1834, normalized to DAPI ($n = 6$ per group). Sham vs UUO, $P < 0.0001$. Bar = 100 μm. Data are presented as mean ± SEM, ****$P < 0.0001$ by t-test. Source data are available online for this figure.

GSE212681

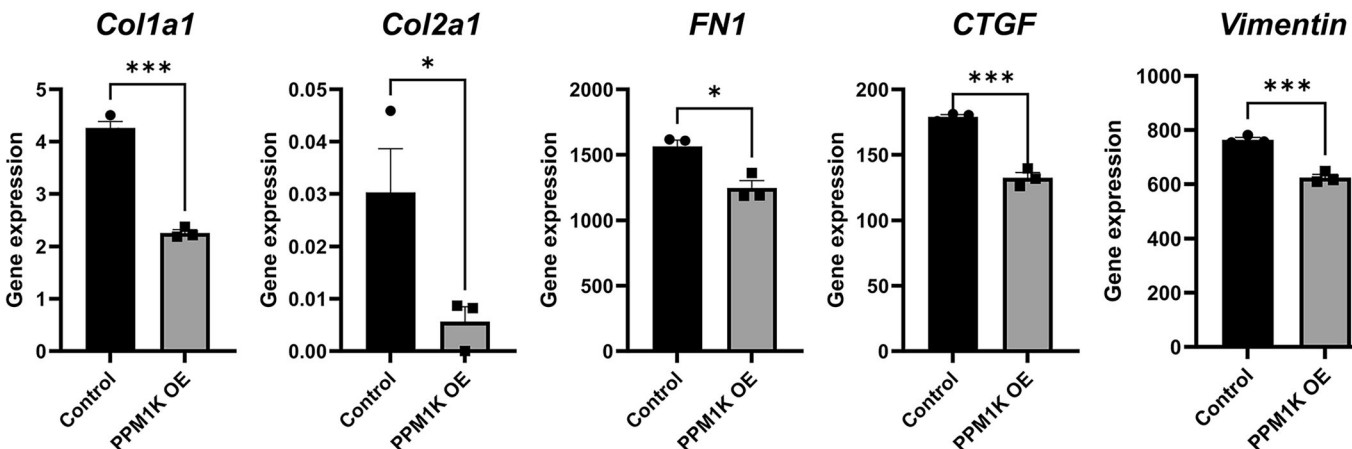

**Figure EV3.** Analysis of fibrosis-related gene expression in PPM1K-overexpressing HK2 cells.

mRNA levels of fibrosis-related genes in PPM1K-overexpressing HK2 cells. RNA-sequencing data retrieved from the Gene Expression Omnibus (GEO) database (GSE212681). *Col1a1*: $P = 0.0001$. *Col2a1*: $P = 0.0492$. *FN1*: $P = 0.0133$. *CTGF*: $P = 0.0004$. *Vimentin*, $P = 0.0007$. Data are presented as mean ± SEM, *$P < 0.01$, **$P < 0.01$, and ***$P < 0.001$ by t-test. Source data are available online for this figure.

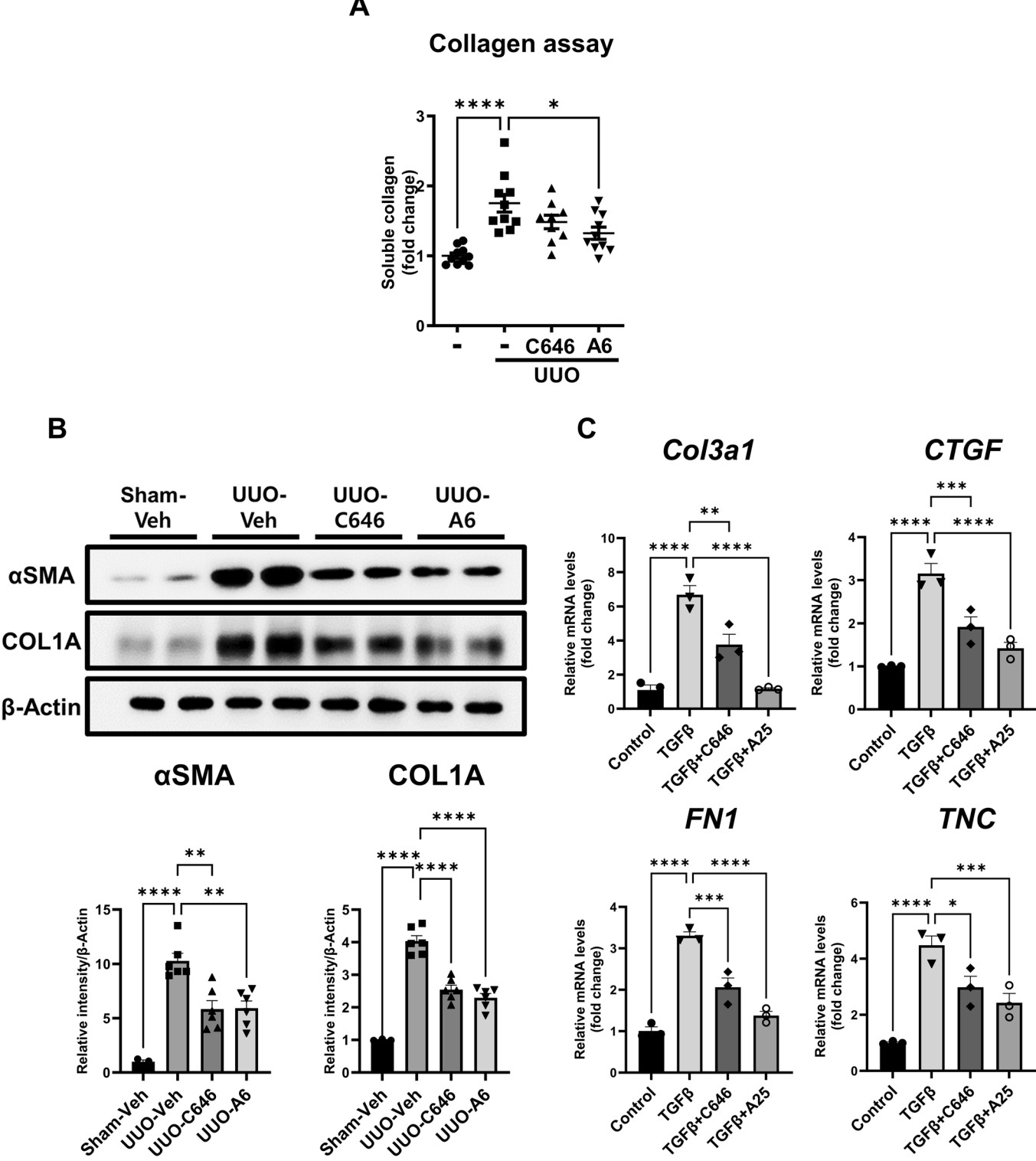

◄ **Figure EV4. Inhibition of p300 suppresses the expression of fibrosis-related marker.**

(A) Soluble collagen assay using kidney samples from UUO-induced fibrosis mice injected with C646 and A6 (sham, $n = 10$, UUO+Veh, $n = 10$, UUO + C646, $n = 9$, UUO + A6, $n = 10$). Sham+Veh vs UUO+Veh, $P < 0.0001$; UUO+Veh vs UUO + A6, $P = 0.0114$. (B) Protein levels of fibrosis-related markers (αSMA and COL1A) in kidney tissues from the UUO-induced fibrosis mouse model injected with C646 and A6. β-Actin was used as the loading control. The graph represents the quantification of intensity in western blot images, normalized to β-Actin (sham, $n = 3$, UUO+Veh, $n = 6$, UUO + C646, $n = 6$, UUO + A6, $n = 6$). αSMA: Sham+Veh vs UUO+Veh, $P < 0.0001$; UUO+Veh vs UUO + C646, $P = 0.0012$; UUO+Veh vs UUO + A6, $P = 0.0015$. COL1A: Sham+Veh vs UUO+Veh, $P < 0.0001$; UUO+Veh vs UUO + C646, $P < 0.0001$; UUO+Veh vs UUO + A6, $P < 0.0001$. (C) mRNA levels of fibrosis-related genes in HK2 cells treated with TGFβ for 24 h and co-treated with C646 and A6 ($n = 3$ per group). *Col3a1*: Control vs TGFβ, $P < 0.0001$; TGFβ vs TGFβ + C646, $P = 0.0011$; TGFβ vs TGFβ + A6, $P < 0.0001$. *CTGF*: Control vs TGFβ, $P < 0.0001$; TGFβ vs TGFβ + C646, $P = 0.0009$; TGFβ vs TGFβ + A6, $P < 0.0001$. *FN1*: Control vs TGFβ, $P < 0.0001$; TGFβ vs TGFβ + C646, $P = 0.0001$; TGFβ vs TGFβ + A6, $P < 0.0001$. *TNC*: Control vs TGFβ, $P < 0.0001$; TGFβ vs TGFβ + C646, $P = 0.0116$; TGFβ vs TGFβ + A6, $P = 0.0009$. Data are presented as mean ± SEM, *$P < 0.05$, **$P < 0.01$, ***$P < 0.001$, and ****$P < 0.0001$ by ordinary one-way ANOVA test. Source data are available online for this figure.

