## [Peer Review File · EMBO Molecular Medicine]

Loss of p300 in proximal tubular cells reduces renal fibrosis and Endothelial-Mesenchymal Transition

Hyunsik Kim, Soo-Yeon Park, Soo Yeon Lee, Jae-Hwan Kwon, Seunghee Byun, Byoungwhi Ko, Jung-Yoon Yoo, Beom seok Kim, Beom Jin Lim, and Ho-Geun Yoon

Corresponding authors: Ho-Geun Yoon (yhgeun@yuhs.ac) , Beom Jin Lim (bjlim@yuhs.ac), Beom seok Kim (DOCBSK@yuhs.ac)

Review Timeline:

Submission Date:	17th Oct 24
Editorial Decision:	6th Nov 24
Appeal:	4th Feb 25
Editorial Decision:	28th Feb 25
Revision Received:	3rd Apr 25
Editorial Decision:	7th Apr 25
Revision Received:	8th Apr 25
Accepted:	9th Apr 25

Editor: Lise Roth

Transaction Report:

6th Nov 2024

Decision on your manuscript EMM-2024-20751

Dear Prof. Yoon,

Thank you for the submission of your manuscript to EMBO Molecular Medicine. We have now received the feedback from the two referees who reviewed your manuscript.

As you will see from the reports below, while they acknowledge the potential interest of the findings, the referees also raise major concerns related to the models used, discrepancies between experiments, low sample numbers and lack of Western Blot quantification.

Based on these concerns and given that at EMBO Press we encourage a single round of revisions in a limited time frame, I am afraid I see little choice but to return the manuscript to you at this point with the decision that we cannot offer to publish it.

I am very sorry to disappoint you in this occasion, and hope that the referees' comments are helpful in your continued work in this area.

Sincerely,

Lise Roth

Lise Roth
Senior Editor
EMBO Molecular Medicine

***** Reviewer's comments *****

Referee #1 (Comments on Novelty/Model System for Author):

The study is of high interest but very difficult to follow. As a nephrology specialist, I do not see the common point between the human (glomerular disease) and experimental data (tubulo-interstitial disease). I have some major issues regarding the UUO model (renal function should not be altered). The in vitro data is excellent!

Referee #1 (Remarks for Author):

This manuscript by Hyunsik Kim et al, describes the role of p300 in proximal tubular cell in renal fibrosis and EndMT. The authors show that this histone-acetyltransferase increased in human biopsies of patients suffering from renal disease, in mice using different models of experimental nephropathy and in tubular cells treated with TGB-b. Knocking down p300, in vivo and in vitro, specifically in proximal renal tubular cells protected against the development of renal fibrosis and consequently this protein may be considered as a promising target against the progression of chronic renal disease.

This is an interesting study with translational perspectives for the treatment of renal disease. The study is

mostly well performed with state-of-the-art methodology. This manuscript however requires significant revision to be more reader friendly and should better highlight the most important findings.

General:

This manuscript contains an enormous amount of data resulting from intensive work. However, the manuscript is poorly organized. In particular, the authors mixed human with experimental in vivo and in vitro data making the story hard to follow. The organization of the manuscript should be extensively revised and significant work should be done to generate a clear picture with a logic that would be easy to follow.

Some major issues:

- I have a major issue concerning measurements of renal function after UO. Given that it is a unilateral ureteral obstruction the other kidney normally should work (hyperfiltration), and thus the renal function should be normal. Please explain the figure 1H. On the other hand, BUU and creatininemia are similar if we compared the UO and the FA models... how this is possible?
- The human data is interesting but mainly concerns glomerular disease, whereas the in vivo experimental data is based on models of tubulointerstitial injury (UO or folic acid). It would be more appropriate to use models of experimental glomerulonephritis.
- The sampling for some in vivo experiments is quite low (n=3). It would be more relevant to focus on one, two models maximum and increased the number of mice.
- In some figures the sham provided pictures look similar with the UO in terms of tubular dilation (Fig1A, D, F and I; FigS1D; FigS2 C, D, E; FigS3 A etc. Please explain.
- The authors provide an enormous quantity of western blots without any quantification.

Minor:

- Please indicate the number of mice used for the study in the legends of figures. Same comment for the in vitro experiments
- In FigS3 B, I do not really see differences between Sham and UNX-STZ kidneys regarding the AQP1+ cells as illustrated in the histograms.

Referee #2 (Comments on Novelty/Model System for Author):

The research technique used in this study is appropriate, and multiple animal models are applied for validation. The study comprehensively explores the role of P300 in renal fibrosis.

Referee #2 (Remarks for Author):

Chronic kidney disease has a high prevalence worldwide. However, its exact pathogenesis and effective treatment methods have not yet been determined. In this study, authors found that loss of p300 in proximal tubular cells reduces renal fibrosis and Endothelial-Mesenchymal Transition. They demonstrate the role of p300 in the development of renal fibrosis, and suggest that p300 is a promising target for treatment of advanced CKD. This topic is interesting, however, some contents should be addressed before published.

Major comments:

- 1.The author should detect whether p300 is expressed in endothelial cells. Because it is reported that endothelial-specific p300 is important in mediating liver fibrosis (PMID: 33159815)
- 2.P300 also serves as a regulator in inflammation, so whether TEC-derived p300 promotes renal fibrosis and EndoMT by regulate inflammation?
- 3.In general, we do not usually evaluate the renal function of UO mouse. Unilateral ligation generally

always has limited impact on renal function.

4. In Figure 2A, TGF- β could significantly promote p300 expression, however, in Figure 2B, TGF- β could not promote p300 expression without MG132 treatment. These two results are contradictory. Moreover, compared Figure 2D and F, baseline of p300 expressions in HK2 cells without treatment are significantly different.

5. Compared Figure 4D and 2F, p300 expressions in HK2 without treatment are significantly different. These results are very confusing.

Minor comments:

1. Please provide the full English name of a proper noun when it first appears, there is no need to repeat it, such as chronic kidney disease.

2. The writing style of proper nouns should be consistent, such as TGF- β and TGF β .

3. Many bands by Western blot assay are overexposed.

4. When abbreviations of proprietary terms first appear, the full name must be provided, such as DM.

5. There are some spelling errors in the article.

It is suggested to use "cKO" to represent conditional p300 knockout from PTC.

As a service to authors, EMBO provides authors with the possibility to transfer a manuscript that one journal cannot offer to publish to another EMBO publication. The full manuscript and if applicable, reviewers reports are automatically sent to the receiving journal to allow for fast handling and a prompt decision on your manuscript. For more details of this service, and to transfer your manuscript to another EMBO title please click on Link Not Available

General Response to Reviewer #1:

We sincerely thank Reviewer #1 for their thoughtful and important comments, as well as their kind recognition of our *in vitro* data. To address your questions regarding human disease and animal models, we have incorporated a new glomerular disease animal model (the Adriamycin model). Additionally, we have done our best to respond to your other detailed questions as comprehensively as possible. Furthermore, we have made substantial revisions to the Results section of the manuscript to enhance readability and have reorganized the Figures to improve clarity for the readers. Once again, we deeply appreciate Reviewer #1's insightful and valuable feedback, which has greatly contributed to improving the quality of our manuscript.

Reviewer 1# Comment 1: I have a major issue concerning measurements of renal function after UUO. Given that it is a unilateral ureteral obstruction the other kidney normally should work (hyperfiltration), and thus the renal function should be normal. Please explain the figure 1H. On the other hand, BUU and creatininemia are similar if we compared the UUO and the FA models... how this is possible?

Answer:

We appreciate the reviewer's insightful comments regarding the renal function measurements in the unilateral ureteral obstruction (UUO) model. As the reviewer pointed out, UUO typically involves ligation of one ureter, with the contralateral kidney compensating through hyperfiltration, which might suggest normal renal function overall. However, following UUO, compensatory hypertrophy of the contralateral kidney due to overload has been reported, along with elevations in blood creatinine, progressive accumulation of inflammatory cells, and glomerular fibrosis¹⁻³. Moreover, numerous studies, including published in *Nature Communications*, have reported elevations in renal function markers, such as BUN and serum creatinine, in UUO models (Panel A)⁴⁻²⁹. Our findings align with these reports, demonstrating that UUO can indeed lead to measurable changes in these markers under certain conditions.

In addition, it has been shown that changes in renal function markers can vary with different experimental conditions in UUO models. For instance, in early UUO models, increases in BUN and serum creatinine have been observed; however, in advanced UUO models, these elevated markers tend to decrease over time (Panel B)^{5,26,27,30}. There is also evidence that age and sex impact renal function marker responses: young rats exhibit elevated markers following UUO (Panel C), whereas older rats do not, and differences in responses have been reported between male and female subjects^{5,30}. Therefore, as the reviewer suggested, factors such as age, sex, and the duration of UUO could result in variability in renal function outcomes, meaning that these markers may not accurately represent overall renal function in the UUO model.

Our results regarding renal function markers likely represent a transient elevation caused by the initial overload on the unilateral kidney in an early UUO model using young mice, which was mitigated by p300 knockout, as indicated in our data. Nonetheless, we agree with the reviewer's observation that renal function markers may be limited as indicators of fibrosis severity and renal function in UUO models.

We sincerely acknowledge and appreciate the reviewers' concerns, and to address these, we have excluded renal function markers as fibrosis evaluation markers in the UUO mouse model from our results to enhance clarity for the readers and prevent any potential misunderstanding.

Additionally, when compared with previous studies published in *Kidney international* and other journals, our results in the FA model demonstrate a similar pattern of elevation in BUN and serum creatinine levels (Panel D)^{26,31-34}. This alignment with earlier reports supports the consistency of our findings with established observations regarding renal function markers in similar experimental models.

Figure for reviewers removed.

Reviewer 1# Comment 2: The human data is interesting but mainly concerns glomerular disease, whereas the in vivo experimental data is based on models of tubulointerstitial injury (UUO or folic acid). It would be more appropriate to use models of experimental glomerulonephritis

Answer: We sincerely appreciate the reviewer 1's comment and wholeheartedly agree that FSGS is primarily a glomerular disease. However, FSGS is also frequently associated with secondary tubulointerstitial damage, inflammation, and fibrosis, which are known to significantly impact disease progression and patient prognosis³⁵⁻⁴². In fact, tubulointerstitial fibrosis is commonly observed alongside glomerular damage in both patients and animal models with FSGS, suggesting that glomerular injury may be linked to or even induce tubulointerstitial lesions^{36,43-46}. In this study, we specifically selected FSGS patient samples exhibiting tubulointerstitial fibrosis to investigate p300 expression in tubular epithelial cells.

Therefore, we employed UUO and folic acid models, which are standard models for studying fibrosis, to directly assess the anti-fibrotic effects. Although these models are not specific to glomerular disease, they effectively mimic common fibrosis pathogenesis, a common feature across various CKD conditions. Therefore, we believe that UUO and folic acid models have value in replicating the common fibrotic processes observed in diseases like advanced FSGS. Moreover, some renal fibrosis studies, including published in *Kidney international*, have used UUO, IRI, and FA models to study fibrosis progression originating from glomerular injuries such as FSGS, IgAN, and LN^{31,47-49}.

In conclusion, fibrosis represents a common final pathway in progressive kidney disease, sharing similar mechanisms regardless of the initial site of injury. Thus, the UUO and folic acid models used in this study are valuable tools for studying fibrosis independently of the specific pathogenesis of each disease. **However, as the reviewer thoughtfully pointed out, it is important to validate the role of p300 in glomerular disease-associated fibrosis. In response to this valuable suggestion, we extended our study by investigating this aspect using p300 knockout mice in the well-established adriamycin-induced glomerular injury model (3 weeks)**⁵⁰⁻⁵².

In this result, we observed a significant reduction in fibrosis in both glomerular and tubular regions within the adriamycin-induced fibrosis model using proximal tubule-specific p300 knockout mice. This finding strongly supports the role of p300 in the general development of fibrosis in kidney diseases. Additionally, we reaffirmed its clinical relevance through further validation in FSGS patients. Therefore, we have added comments related to these results in the manuscript. (Page 8, Line 7 and Fig. S8A-E)

Figures for reviewers removed

Reviewer 1# Comment 3: The sampling for some in vivo experiments is quite low (n=3). It would be more relevant to focus on one, two models maximum and increased the number of mice.

Answer: We greatly appreciate the reviewer's thoughtful concern regarding the sample size in some of the in vivo experiments. We initially considered that a sample size of n=3 would be sufficient to achieve statistical significance for these proof-of-concept experiments. However, we understand the reviewer's concern that the absolute sample size might be perceived as limited. In response, we have reanalyzed the data by incorporating additional samples that were already available, ensuring a more robust statistical evaluation. We sincerely thank the reviewer for highlighting this point, as it has allowed us to strengthen the conclusions of our study. **(Fig. 1B, Fig. 1D, Fig. 5G, and Fig. S1E have been revised to incorporate additional samples)**

Reviewer 1# Comment 4: In some figures the sham provided pictures look similar with the UUO in terms of tubular dilation (Fig1A, D, F and I; FigS1D; FigS2 C, D, E; FigS3 A etc. Please explain

Answer: The renal tissue has a three-dimensional tubular structure, which can appear different depending on the orientation (vertical or horizontal) of the tissue section obtained after processes such as tissue flushing, fixation, and preparation, resulting in variations in appearance in certain images. However, these variations do not significantly affect the assessment of ECM accumulation or protein expression in our fibrosis-related studies. Other published results have also shown empty spaces in the tubular regions of sham kidneys (Panel A-C)⁵³⁻⁵⁷. While UUO is indeed characterized by tubular dilation, as the reviewer mentioned, it is also accompanied by ECM accumulation in the interstitium. Our sham images are distinct from UUO images in this regard, as they do not exhibit the same degree of ECM accumulation. For reference, we have included our low-magnification H&E images of an untreated kidney from the same mouse, along with the UUO-treated mouse kidney, to highlight these differences (Panel D). Additionally, as for Figure 1A, it is the result of a commercially available normal human sample slide. However, as the reviewer noted, the current images may cause confusion for readers. Therefore, some figures showing excessively large empty spaces in tubular regions have been replaced by substituting the images or re-sectioning the paraffin tissue samples currently available. **(Fig. 1A, Fig. 1D, Fig. 1F, Fig. 6D, Fig. S3D, Fig. S3E, Fig. S4A and Fig. S6D have been revised)**

Figures for reviewers removed

Figure for reviewers removed

Reviewer 1# Comment 5: The authors provide an enormous quantity of western blots without any quantification.

Answer: We sincerely appreciate the reviewer's feedback regarding the western blots. In the revised manuscript, we have provided quantification of the western blot results alongside the images to enhance clarity and reproducibility. However, we respectfully note that quantification was not included for the condition-specific experiments and immunoprecipitation (IP) experiments, as these were intended to be qualitative assessments. We thank the reviewer for their valuable input, which has allowed us to improve the presentation of our data. (Quantification graphs have been added to Fig. 2C, Fig. 2D, Fig. 2E, Fig. 3E, Fig. 4D, Fig. S6C, Fig. S7A, Fig. S8D, Fig. S9A, Fig. S9C, Fig. S10C, Fig. S10D, Fig. S12C, Fig. S13A, Fig. S14D, Fig. S14G, Fig. S16A, Fig. S16C, Fig. S23B, Fig. S26B, and Fig. S27B.)

Reviewer 1# Minor comments 1: Please indicate the number of mice used for the study in the legends of figures. Same comment for the in vitro experiments.

Answer: Thank you for this important suggestion. We have now indicated the number of mice used for each study within the figure legends and included the sample sizes for in vitro experiments to ensure transparency and reproducibility.

Reviewer 1# Minor comments 2: In Fig S3B, I do not really see differences between Sham and UNX-STZ kidneys regarding the AQP1+ cells as illustrated in the histograms.

Answer: We understand the reviewer's concern regarding the difficulty in distinguishing differences between Sham and UNX-STZ kidneys in AQP1+ cells as presented in the histograms of Figure S3B. To address this, we have adjusted the image contrast and modified the scale of data presentation in this figure to better clarify the observed differences and ensure accurate representation. **(Fig. S3B have been revised)**

Figure for reviewers removed

General Response to Reviewer #2:

We sincerely thank Reviewer #2 for their detailed comments and insightful questions regarding the p300 protein. In responding to these questions, we gained valuable insights into the role of endothelial p300 in the progression of fibrosis, as well as the potential association of p300 in proximal tubular cells with inflammation. Furthermore, we deeply appreciate the reviewer's meticulous attention to minor textual errors in the manuscript, which allowed us to refine its quality and improve its reader-friendliness. Once again, we are truly grateful for Reviewer #2's invaluable feedback, which has significantly contributed to enhancing the overall quality of our manuscript.

Reviewer 2# Comment 1: The author should detect whether p300 is expressed in endothelial cells. Because it is reported that endothelial-specific p300 is important in mediating liver fibrosis.

Answer: Thank you for the valuable insight. As the reviewer pointed out, p300 levels in endothelial cells are indeed elevated in liver fibrosis and are known to contribute to fibrogenesis. Additionally, **it has been reported that p300 expression in endothelial cells increases under high glucose conditions**, which is a major risk factor for chronic kidney disease (CKD) (Panel A)⁵⁸. Consistent with these findings, in Fig. 1C, we evaluated the expression of p300 in CD31-positive cells (endothelial cells) in the UUO-induced kidney fibrosis mouse model (Panel B). While the increase in p300 expression was not statistically significant, a slight elevation was observed.

However, among the various cell types in the kidney, proximal tubular cells constitute a major proportion, and the magnitude of p300 increase was most prominent in these cells. Therefore, we focused on p300 in proximal tubular cells. **Additionally, in the UUO (8-day) mouse model, we confirmed through immunofluorescence (IF) staining that p300 expression slightly increased (1.5-fold change) in CD31+ cells (endothelial cells) (Panel C).** Although this increase was modest compared to that in proximal tubular cells, it suggests the possibility that, as reported in other organs, increased p300 in vascular endothelial cells within the kidney may also contribute to fibrosis progression.

In accordance with the reviewer's advice, we have emphasized in the Discussion the need for further studies to investigate the role of p300 in endothelial cells during kidney fibrosis. (Page 21, Line 10) Specifically, future research should aim to determine the physiological conditions that drive p300 level increases in endothelial cells within the kidney, elucidate the molecular mechanisms regulating these changes, and clarify their contribution to fibrosis progression.

Figures for reviewers removed

Reviewer 2# Comment 2: p300 also serves as a regulator in inflammation, so whether TEC-derived p300 promotes renal fibrosis and EndoMT by regulate inflammation?

Answer: We appreciate the reviewer's insightful questions and comments. As noted, it is well established that p300 plays a crucial role in regulating inflammation.

Our findings in Fig. 4C demonstrate that proximal tubule-specific p300 knockout (KO) affects various biological pathways (Panel A). In this analysis, changes in inflammation-related pathways did not rank among the top pathways regulated by proximal tubule-specific p300 KO. However, as the reviewer pointed out, RNA-seq results revealed that inflammation-related genes upregulated by UUO were downregulated in the absence of p300 (Panel B, red box). Additionally, we observed a reduction in key inflammatory markers, including interleukins, CXCL, and CCL family genes (Panel C). **These results suggest that p300 in proximal tubular cells (PTCs) plays a role in regulating some inflammation-related genes under fibrotic conditions, but its impact appears to be minor compared to changes in other biological pathways.**

Inflammation is a major driver of fibrosis and has been implicated in regulating processes such as endothelial-to-mesenchymal transition (EndoMT). However, it remains unclear whether fibrosis and EndoMT are directly regulated through inflammatory signaling, emphasizing the need for further in-depth investigation.

Figures for reviewers removed

Reviewer 2# Comment 3: In general, we do not usually evaluate the renal function of UUO mouse. Unilateral ligation generally always has limited impact on renal function.

Answer: We sincerely appreciate the reviewer's comment regarding the evaluation of renal function in UUO models. As both reviewer 1# and reviewer 2# correctly noted, unilateral ligation in UUO generally results in a minimal impact on overall renal function due to compensatory hyperfiltration by the contralateral kidney. However, numerous studies have reported that UUO can still lead to measurable changes in markers such as BUN and serum creatinine, particularly in early stages or under specific experimental conditions¹⁻²⁹. These changes have been attributed to transient functional overload, inflammatory responses, and fibrotic processes affecting the obstructed kidney.

Our findings are consistent with these reports, as we observed a slight but measurable elevation in renal function markers in the UUO model, which was mitigated by p300 knockout. Nevertheless, we fully agree with the reviewer that these markers may have limitations in reflecting renal function or fibrosis severity in UUO models. **To address this concern, we have excluded renal function markers as indicators of fibrosis or renal function in our revised results for the UUO model to avoid any potential misunderstanding.**

Once again, we thank the reviewer for highlighting this important point and for their valuable feedback, which has allowed us to clarify and improve the presentation of our findings.

Reviewer 2# Comment 4: In Figure 2A, TGF- β could significantly promote p300 expression, however, in Figure 2B, TGF- β could not promote p300 expression without MG132 treatment. These two results are contradictory. Moreover, compared Figure 2D and F, baseline of p300 expressions in HK2 cells without treatment are significantly different.

Answer: We sincerely thank the reviewer for their thoughtful observation regarding the apparent differences in p300 expression between Figures 2A and 2D. As the reviewer has correctly pointed out, there appears to be a significant difference in p300 expression. **We would like to clarify that this difference can be attributed to experimental factors, particularly differences in cell types and treatment durations.**

First, the two experiments were conducted using different cell types: mouse primary proximal tubular cells (Figure 2A) and the HK2 cell line (Figure 2D). Second, there is a notable difference in the treatment durations between the two experiments. In Figure 2A, p300 expression was assessed after **12 and 24 hours** of TGF- β treatment to evaluate its induction over a longer time frame. In contrast, Figure 2D was designed to assess the stabilization of p300 protein by TGF- β in the presence of MG132, with the treatment duration **limited to 6 hours**. This time point was selected based on the results shown in Fig. S6C (Panel A), which demonstrated that MG132 treatment led to the most significant increase in p300 expression at 6 hours. Consequently, the increase in p300 expression from TGF- β treatment alone appeared minimal in Figure 2D. **To address this, we have now clarified the treatment durations of TGF- β and MG132 in the figure legends.** Additionally, in the uncropped blot image of Figure 2D (Panel B), we confirmed that the p300 expression pattern after 6 hours of TGF- β treatment alone was consistent with the results shown in Fig. S6C. **To further improve clarity, we have replaced the original image with one with improved contrast.**

Furthermore, the apparent differences in baseline p300 levels between Figures 2D (Panel C) and 2F (Panel D) can also be attributed to the experimental conditions. In Figure 2D, MG132 was used to stabilize cellular proteins, leading to excessive accumulation of p300. **If the same exposure times as Figure 2F were applied, the MG132-treated bands would have been overly saturated. To avoid this, we used shorter exposure times in Figure 2D, which may have caused the baseline p300 levels to appear different.**

We sincerely hope that these adjustments and clarifications adequately address the reviewer's concerns and improve the clarity and accuracy of the presented data

Figures for reviewers removed

Reviewer 2# Comment 5: Compared Figure 4D and 2F, p300 expressions in HK2 without treatment are significantly different. These results are very confusing.

Answer: We sincerely acknowledge the reviewer's concern and greatly appreciate the insightful comments. As the reviewer pointed out, the baseline levels of p300 in Figure 4D may appear lower compared to Figure 2F. However, the experiment in Figure 4D was specifically designed to examine changes in p300 expression over a 24-hour period following TGF- β treatment. To avoid oversaturation of the p300 band at the 24-hour time point, a shorter exposure time was necessary during band detection. Consequently, a lower exposure time was used for Figure 4D compared to Figure 2F.

Nonetheless, to address any potential concerns regarding the baseline levels of p300 in Figure 4D, we reloaded the same samples used in the previous experiment with double the protein amount. Upon comparison, the fold increase in p300 expression after TGF- β treatment at the same time point (6 hours) was found to be consistent between the two experiments, showing fold changes of 1.51 and 1.54, respectively. Additionally, when Figure 4D was intentionally overexposed, the band intensity appeared comparable to that in Figure 2F (Panel A).

To improve clarity, we have replaced the p300 image in Figure 4D with the reloaded samples. While the baseline differences in p300 levels between Figures 4D and 2F can be attributed to the experimental conditions, we believe that these differences are acceptable and consistent with the context of the respective experiments (**Fig. 4D p300 band image was revised**). We hope that the explanation of these experimental conditions and the implemented revisions have adequately addressed the reviewer's concerns.

Figures for reviewers removed

Reviewer 2# Minor comments 1: Please provide the full English name of a proper noun when it first appears, there is no need to repeat it, such as chronic kidney disease.

Answer: Thank you for this valuable feedback. We have ensured that the full English names of proper nouns, such as "chronic kidney disease," are provided upon their first mention and have avoided unnecessary repetition throughout the manuscript.

Reviewer 2# Minor comments 2: The writing style of proper nouns should be consistent, such as TGF- β and TGF β .

Answer: We sincerely appreciate the feedback on maintaining consistency in the writing style of proper nouns. We have standardized all terms, ensuring that "TGF β " is used consistently throughout the manuscript.

Reviewer 2# Minor comments 3: Many bands by Western blot assay are overexposed.

Answer: We acknowledge that some Western blot bands appear overexposed. To address this, we have adjusted the exposure times and reprocessed these images as necessary to improve their clarity and accuracy.

Reviewer 2# Minor comments 4: When abbreviations of proprietary terms first appear, the full name must be provided, such as DM.

Answer: Thank you for pointing this out. We have ensured that abbreviations, such as "DN," are accompanied by their full names (e.g., "diabetic nephropathy") upon their first mention in the text for clarity.

Reviewer 2# Minor comments 5: There are some spelling errors in the article. It is suggested to use "cKO" to represent conditional p300 knockout from PTC.

Answer: We thoroughly proofread the manuscript to correct any spelling errors. Additionally, we appreciated the suggestion to use "p300 cKO" to represent conditional p300 knockout from PTC and implemented this notation for clarity and consistency.

Reference

1. Zhang L, Mo X, Jiang Z, et al. Contralateral renal change in a unilateral ureteral obstruction rat model using intravoxel incoherent motion diffusion-weighted imaging. *Ren Fail.* Dec 2024;46(2):2359642. doi:10.1080/0886022X.2024.2359642
2. Figueroa SM, Lozano M, Lobos C, Hennrikus MT, Gonzalez AA, Amador CA. Upregulation of Cortical Renin and Downregulation of Medullary (Pro)Renin Receptor in Unilateral Ureteral Obstruction. *Front Pharmacol.* 2019;10:1314. doi:10.3389/fphar.2019.01314
3. Xiong Y, Chang Y, Hao J, et al. Eplerenone Attenuates Fibrosis in the Contralateral Kidney of UO Rats by Preventing Macrophage-to-Myofibroblast Transition. *Front Pharmacol.* 2021;12:620433. doi:10.3389/fphar.2021.620433
4. Zhou W, Wu WH, Si ZL, et al. The gut microbe *Bacteroides fragilis* ameliorates renal fibrosis in mice. *Nat Commun.* Oct 14 2022;13(1):6081. doi:10.1038/s41467-022-33824-6
5. Munguia-Galaviz FJ, Miranda-Diaz AG, Gutierrez-Mercado YK, et al. The Sigma-1 Receptor Exacerbates Cardiac Dysfunction Induced by Obstructive Nephropathy: A Role for Sexual Dimorphism. *Biomedicines.* Aug 20 2024;12(8)doi:10.3390/biomedicines12081908
6. Liao Y, Tan RZ, Li JC, et al. Isoliquiritigenin Attenuates UUO-Induced Renal Inflammation and Fibrosis by Inhibiting Mincle/Syk/NF-Kappa B Signaling Pathway. *Drug Des Devel Ther.* 2020;14:1455-1468. doi:10.2147/DDDT.S243420
7. Ham O, Jin W, Lei L, et al. Pathological cardiac remodeling occurs early in CKD mice from unilateral urinary obstruction, and is attenuated by Enalapril. *Sci Rep.* Oct 31 2018;8(1):16087. doi:10.1038/s41598-018-34216-x
8. Zhao J, Meng M, Zhang J, et al. Astaxanthin ameliorates renal interstitial fibrosis and peritubular capillary rarefaction in unilateral ureteral obstruction. *Mol Med Rep.* Apr 2019;19(4):3168-3178. doi:10.3892/mmr.2019.9970
9. Wu J, Xu Y, Geng Z, et al. Chitosan oligosaccharide alleviates renal fibrosis through reducing oxidative stress damage and regulating TGF-beta1/Smads pathway. *Sci Rep.* Nov 10 2022;12(1):19160. doi:10.1038/s41598-022-20719-1
10. Park JH, Leem J, Lee SJ. Protective Effects of Carnosol on Renal Interstitial Fibrosis in a Murine Model of Unilateral Ureteral Obstruction. *Antioxidants (Basel).* Nov 26 2022;11(12)doi:10.3390/antiox11122341
11. Huang H, Liu Q, Zhang T, et al. Farnesylthiosalicylic Acid-Loaded Albumin Nanoparticle Alleviates Renal Fibrosis by Inhibiting Ras/Raf1/p38 Signaling Pathway. *Int J Nanomedicine.* 2021;16:6441-6453. doi:10.2147/IJN.S318124
12. Wang J, Ge S, Wang Y, et al. Puerarin Alleviates UUO-Induced Inflammation and Fibrosis by Regulating the NF-kappaB P65/STAT3 and TGFbeta1/Smads Signaling Pathways. *Drug Des Devel Ther.* 2021;15:3697-3708. doi:10.2147/DDDT.S321879
13. Sun J, Zhang S, Shi B, Zheng D, Shi J. Transcriptome Identified lncRNAs Associated with Renal Fibrosis in UUO Rat Model. *Front Physiol.* 2017;8:658. doi:10.3389/fphys.2017.00658
14. Zou J, Zhou X, Chen X, Ma Y, Yu R. Shenkang Injection for Treating Renal Fibrosis-Metabonomics and Regulation of E3 Ubiquitin Ligase Smurfs on TGF-beta/Smads Signal Transduction. *Front Pharmacol.* 2022;13:849832. doi:10.3389/fphar.2022.849832
15. Zhao D, Luan Z. Oleonic Acid Attenuates Renal Fibrosis through TGF-beta/Smad Pathway in a Rat Model of Unilateral Ureteral Obstruction. *Evid Based Complement Alternat Med.* 2020;2020:2085303. doi:10.1155/2020/2085303
16. Sun WB, Liu HJ, Wu BQ, Dai LM, Ren Y, Zheng DN. Uncovering the Nephroprotective Mechanism of Caffeic Acid in Renal Tubulointerstitial Fibrosis through Network Pharmacology Analysis. *Nat Prod Commun.* Mar 2024;19(3). doi:1934578x241237894 10.1177/1934578x241237894
17. Shen W, Fan K, Zhao Y, Zhang J, Xie M. Stevioside inhibits unilateral ureteral obstruction-induced kidney fibrosis and upregulates renal PPARgamma expression in mice. *J Food Biochem.* Dec 2020;44(12):e13520. doi:10.1111/jfbc.13520
18. Yuan X, Zhang J, Xie F, et al. Loss of the Protein Cystathionine beta-Synthase During Kidney Injury Promotes Renal Tubulointerstitial Fibrosis. *Kidney Blood Press Res.* 2017;42(3):428-443. doi:10.1159/000479295
19. Lai J, Huang L, Bao Y, et al. A deep clustering-based mass spectral data visualization strategy for anti-renal fibrotic lead compound identification from natural products. *Analyst.* Oct 24 2022;147(21):4739-4751. doi:10.1039/d2an01185a
20. Hu X, Yang M, Li X, Gong Z, Duan J. Myo-Inositol Attenuates Renal Interstitial Fibrosis in Obstructive Nephropathy by Inhibiting PI3K/AKT Activation. *J Med Food.* Jun 2023;26(6):368-378. doi:10.1089/jmf.2022.K.0152
21. Zhang ZH, He JQ, Zhao YY, Chen HC, Tan NH. Asiatic acid prevents renal fibrosis in UUO rats via promoting the production of 15d-PGJ2, an endogenous ligand of PPAR-gamma. *Acta Pharmacol Sin.* Mar 2020;41(3):373-382. doi:10.1038/s41401-019-0319-4
22. Chen Z, Wu S, Huang L, et al. Colonic microflora and plasma metabolite-based comparative analysis of unilateral ureteral obstruction-induced chronic kidney disease after treatment with the Chinese medicine FuZhengHuaYuJiangZhuTongLuo and AST-120. *Heliyon.* Feb 15 2024;10(3):e24987. doi:10.1016/j.heliyon.2024.e24987
23. Hsieh YH, Tsai JP, Ting YH, Hung TW, Chao WW. Rosmarinic acid ameliorates renal interstitial fibrosis by inhibiting the phosphorylated-AKT mediated epithelial-mesenchymal transition in vitro and in vivo. *Food Funct.* Apr 20 2022;13(8):4641-4652. doi:10.1039/d2fo00204c
24. Zhou Y, Zhu X, Wang X, et al. H(2)S alleviates renal injury and fibrosis in response to unilateral ureteral obstruction by regulating macrophage infiltration via inhibition of NLRP3 signaling. *Exp Cell Res.* Feb 1 2020;387(1):111779. doi:10.1016/j.yexcr.2019.111779
25. Lu S, Fan HW, Li K, Fan XD. Suppression of Elp2 prevents renal fibrosis and inflammation induced by unilateral ureter obstruction (UUO) via inactivating Stat3-regulated TGF-beta1 and NF-kappaB pathways. *Biochem Biophys Res Commun.* Jun 22 2018;501(2):400-407. doi:10.1016/j.bbrc.2018.04.227

26. Price NL, Miguel V, Ding W, et al. Genetic deficiency or pharmacological inhibition of miR-33 protects from kidney fibrosis. *JCI Insight*. Nov 14 2019;4(22)doi:10.1172/jci.insight.131102
27. Belghasem ME, A'Amar O, Roth D, et al. Towards minimally-invasive, quantitative assessment of chronic kidney disease using optical spectroscopy. *Sci Rep*. May 9 2019;9(1):7168. doi:10.1038/s41598-019-43684-8
28. Jin Y, Shao X, Sun B, Miao C, Li Z, Shi Y. Urinary kidney injury molecule-1 as an early diagnostic biomarker of obstructive acute kidney injury and development of a rapid detection method. *Mol Med Rep*. Mar 2017;15(3):1229-1235. doi:10.3892/mmr.2017.6103
29. Wang L, Ma J, Guo C, et al. Danggui Buxue Tang Attenuates Tubulointerstitial Fibrosis via Suppressing NLRP3 Inflammasome in a Rat Model of Unilateral Ureteral Obstruction. *Biomed Res Int*. 2016;2016:9368483. doi:10.1155/2016/9368483
30. Wang Y, Guo YF, Fu GP, et al. Protective effect of miRNA-containing extracellular vesicles derived from mesenchymal stromal cells of old rats on renal function in chronic kidney disease. *Stem Cell Res Ther*. Jul 8 2020;11(1):274. doi:10.1186/s13287-020-01792-7
31. Shi Y, Tao M, Chen H, et al. Ubiquitin-specific protease 11 promotes partial epithelial-to-mesenchymal transition by deubiquitinating the epidermal growth factor receptor during kidney fibrosis. *Kidney Int*. Mar 2023;103(3):544-564. doi:10.1016/j.kint.2022.11.027
32. Doi K, Leelahavanichkul A, Hu X, et al. Pre-existing renal disease promotes sepsis-induced acute kidney injury and worsens outcome. *Kidney Int*. Oct 2008;74(8):1017-25. doi:10.1038/ki.2008.346
33. Jin H, Yang Y, Zhu X, et al. DDRGK1-mediated ER-phagy attenuates acute kidney injury through ER-stress and apoptosis. *Cell Death Dis*. Jan 17 2024;15(1):63. doi:10.1038/s41419-024-06449-4
34. Li X, Zou Y, Fu YY, et al. A-Lipoic Acid Alleviates Folic Acid-Induced Renal Damage Through Inhibition of Ferroptosis. *Front Physiol*. 2021;12:680544. doi:10.3389/fphys.2021.680544
35. De Vriese AS, Sethi S, Nath KA, Glassock RJ, Fervenza FC. Differentiating Primary, Genetic, and Secondary FSGS in Adults: A Clinicopathologic Approach. *J Am Soc Nephrol*. Mar 2018;29(3):759-774. doi:10.1681/ASN.2017090958
36. Kriz W, Hosser H, Hahnel B, Gretz N, Provoost AP. From segmental glomerulosclerosis to total nephron degeneration and interstitial fibrosis: a histopathological study in rat models and human glomerulopathies. *Nephrol Dial Transplant*. Nov 1998;13(11):2781-98. doi:10.1093/ndt/13.11.2781
37. Tuttle KR, Abner CW, Walker PD, et al. Clinical Characteristics and Histopathology in Adults With Focal Segmental Glomerulosclerosis. *Kidney Med*. Feb 2024;6(2):100748. doi:10.1016/j.xkme.2023.100748
38. Alexopoulos E, Stangou M, Papagianni A, Pantzaki A, Papadimitriou M. Factors influencing the course and the response to treatment in primary focal segmental glomerulosclerosis. *Nephrol Dial Transplant*. Sep 2000;15(9):1348-56. doi:10.1093/ndt/15.9.1348
39. Futrakul P, Yenrudi S, Futrakul N, et al. Tubular function and tubulointerstitial disease. *Am J Kidney Dis*. May 1999;33(5):886-91. doi:10.1016/s0272-6386(99)70421-x
40. Reidy K, Kaskel FJ. Pathophysiology of focal segmental glomerulosclerosis. *Pediatr Nephrol*. Mar 2007;22(3):350-4. doi:10.1007/s00467-006-0357-2
41. Lim BJ, Yang JW, Zou J, et al. Tubulointerstitial fibrosis can sensitize the kidney to subsequent glomerular injury. *Kidney Int*. Dec 2017;92(6):1395-1403. doi:10.1016/j.kint.2017.04.010
42. Sun K, Xie Q, Hao CM. Mechanisms of Scarring in Focal Segmental Glomerulosclerosis. *Kidney Dis (Basel)*. Sep 2021;7(5):350-358. doi:10.1159/000517108
43. Fan Y, Dong S, Xia Y, et al. Role of TSP-1 and its receptor ITGB3 in the renal tubulointerstitial injury of focal segmental glomerulosclerosis. *J Biol Chem*. Aug 2024;300(8):107516. doi:10.1016/j.jbc.2024.107516
44. Wilkening A, Krappe J, Muhe AM, et al. C-C chemokine receptor type 2 mediates glomerular injury and interstitial fibrosis in focal segmental glomerulosclerosis. *Nephrol Dial Transplant*. Feb 1 2020;35(2):227-239. doi:10.1093/ndt/gfy380
45. Lin YC, Hwu Y, Huang GS, et al. Differential synchrotron X-ray imaging markers based on the renal microvasculature for tubulointerstitial lesions and glomerulopathy. *Sci Rep*. Jun 14 2017;7(1):3488. doi:10.1038/s41598-017-03677-x
46. Kokeny G, Nemeth A, Kopp JB, et al. Susceptibility to kidney fibrosis in mice is associated with early growth response-2 protein and tissue inhibitor of metalloproteinase-1 expression. *Kidney Int*. Aug 2022;102(2):337-354. doi:10.1016/j.kint.2022.03.029
47. Wang D, Li Y, Li G, et al. Inhibition of PKC-delta retards kidney fibrosis via inhibiting cGAS-STING signaling pathway in mice. *Cell Death Discov*. Jul 7 2024;10(1):314. doi:10.1038/s41420-024-02087-z
48. Chen B, Wang P, Liang X, et al. Permissive effect of GSK3beta on profibrogenic plasticity of renal tubular cells in progressive chronic kidney disease. *Cell Death Dis*. Apr 30 2021;12(5):432. doi:10.1038/s41419-021-03709-5
49. Raza S, Jokl E, Pritchett J, et al. SOX9 is required for kidney fibrosis and activates NAV3 to drive renal myofibroblast function. *Sci Signal*. Mar 2 2021;14(672)doi:10.1126/scisignal.abb4282
50. Fogo AB. Animal models of FSGS: lessons for pathogenesis and treatment. *Semin Nephrol*. Mar 2003;23(2):161-71. doi:10.1053/snep.2003.50015
51. Yang HC, Zuo Y, Fogo AB. Models of chronic kidney disease. *Drug Discov Today Dis Models*. 2010;7(1-2):13-19. doi:10.1016/j.ddmod.2010.08.002
52. Yang JW, Dettmar AK, Kronbichler A, et al. Recent advances of animal model of focal segmental glomerulosclerosis. *Clin Exp Nephrol*. Aug 2018;22(4):752-763. doi:10.1007/s10157-018-1552-8
53. Hong Q, Cai H, Zhang L, et al. Modulation of transforming growth factor-beta-induced kidney fibrosis by leucine-rich α -2 glycoprotein-1. *Kidney Int*. Feb 2022;101(2):299-314. doi:10.1016/j.kint.2021.10.023
54. Song K, Wang F, Li Q, et al. Hydrogen sulfide inhibits the renal fibrosis of obstructive nephropathy. *Kidney Int*.

Jun 2014;85(6):1318-29. doi:10.1038/ki.2013.449

55. Wang Y, Wang Y, Li Y, et al. Metformin attenuates renal interstitial fibrosis through upregulation of Deptor in unilateral ureteral obstruction in rats. *Exp Ther Med*. Nov 2020;20(5):17. doi:10.3892/etm.2020.9144

56. Abou Taha MA, Ali FEM, Saleh IG, Akool ES. Sorafenib and edaravone protect against renal fibrosis induced by unilateral ureteral obstruction via inhibition of oxidative stress, inflammation, and RIPK-3/MLKL pathway. *Naunyn Schmiedebergs Arch Pharmacol*. Nov 2024;397(11):8961-8977. doi:10.1007/s00210-024-03146-z

57. Tubular insulin-induced gene 1 deficiency promotes NAD(+) consumption and exacerbates kidney fibrosis. *EMBO Mol Med*. Jul;16(7):1675-1703. doi: 10.1038/s44321-024-00081-7.

58. Chen S, Feng B, George B, Chakrabarti R, Chen M, Chakrabarti S. Transcriptional coactivator p300 regulates glucose-induced gene expression in endothelial cells. *Am J Physiol Endocrinol Metab*. Jan 2010;298(1):E127-37. doi:10.1152/ajpendo.00432.2009

28th Feb 2025

Dear Prof. Yoon,

Thank you for submitting your revised study. We have now received the reports from the two referees who had also evaluated your initial submission. As you will see from the reports below, they are satisfied with the revisions, and I will therefore be able to accept your manuscript once the following editorial issues are addressed:

1/ Manuscript text:

- Please provide your manuscript text as a word document and indicate in track changes mode any new modification.
- All Materials and Methods need to be described in the main text using our 'Structured Methods' format. According to this format, the Methods section includes a Reagents and Tools Table (listing key reagents, experimental models, software and relevant equipment and including their sources and relevant identifiers) followed by a Methods and Protocols section describing the methods, ideally using a step-by-step protocol format. The aim is to facilitate adoption of the methodologies across labs. Please download and fill our Reagents and Tools Table template (.docx), which you can find in our author guidelines: <https://www.embopress.org/page/journal/14693178/authorguide#structuredmethods>. When submitting your revised manuscript, please do not include the Reagents and Tools Table in the Methods section of the manuscript but upload it as a separate file choosing the file type "Reagent Table". An example of a Method paper with Structured Methods can be found here: <https://www.embopress.org/doi/10.15252/msb.20178071>
- "Materials and Methods" should be renamed "Methods".
- Data availability: Please provide links to access the data, and kindly note that the datasets must be publicly accessible before acceptance of the manuscript. Please remove: "All other data supporting the findings of this study are available upon reasonable request."
- Author contributions: CRediT has replaced the traditional author contributions section because it offers a systematic machine readable author contributions format that allows for more effective research assessment. Please remove the Authors Contributions from the manuscript and use the free text boxes beneath each contributing author's name in our system to add specific details on the author's contribution. More information is available in our guide to authors.
- Please rename "Declaration of interests" to "Disclosure and competing interests".
- Please correct the reference format to alphabetical order, with 10 author names listed before et. al. DOIs should be removed.

2/ Figures, EV figures, and Appendix:

- The main figures should be removed from the manuscript text and uploaded as individual, high resolution figure files. The legends should be compiled at the end of the manuscript text.
- The section with the supplementary information should be removed from the manuscript text. Please note that we replaced Supplementary Information with Expanded View (EV) Figures and Tables that are collapsible/expandable online. EV Figures should be cited as 'Figure EV1, Figure EV2' etc... in the text and their respective legends should be included in the main text after the legends of regular figures. For the figures that you do NOT wish to display as Expanded View figures, they should be bundled together with their legends in a single PDF file called *Appendix*, which should start with a short Table of Content. Appendix figures should be referred to in the main text as: "Appendix Figure S1, Appendix Figure S2" etc.

Additional Tables/Datasets should be labeled and referred to as Table EV1, Dataset EV1, etc. Legends have to be provided in a separate tab in case of .xls files. Alternatively, the legend can be supplied as a separate text file (README) and zipped together with the Table/Dataset file.

- There is a callout for a Table 1 (antibody information) that doesn't exist, please correct.
- Please carefully check the composition of Fig S8A and correct if needed. Please note that re-use of an image should be indicated in the figure legend.
- Please address the queries from our copy editors:
 1. Please note that the legend for figure 5F is missing in the manuscript. This needs to be rectified.
 2. Please note that the figure 5E is missing from the manuscript. This needs to be rectified.
 3. Please note that the exact p values are not provided in the legends of figures 1A-I; 2A-I; 3A, B, D, E, F, G; 4D, F, G; 5A-D; 6A-D..

3/ Please provide the requested Source Data as 1 file per figure.

4/ Please provide a complete author checklist, which you can download from our author guidelines (<https://www.embopress.org/page/journal/17574684/authorguide#submissionofrevisions>). Please insert information in the checklist that is also reflected in the manuscript. The completed author checklist will also be part of the RPF.

5/ Please note that all corresponding authors are required to supply an ORCID ID for their name upon submission of a revised manuscript. An ORCID identifier is still missing for Beom Jin Lim.

6/ Please provide 'The paper explained': EMBO Molecular Medicine articles are accompanied by a summary of the articles to emphasize the major findings in the paper and their medical implications for the non-specialist reader. Please provide a draft summary of your article highlighting

7/ Every published paper includes a 'Synopsis' to further enhance discoverability. Synopses are displayed on the journal webpage and are freely accessible to all readers. They include a short stand first (maximum of 300 characters, including space) as well as 2-5 one-sentences bullet points that summarizes the paper. Please write the bullet points to summarize the key NEW findings. They should be designed to be complementary to the abstract - i.e. not repeat the same text. We encourage inclusion of key acronyms and quantitative information (maximum of 30 words / bullet point). Please use the passive voice. Please attach these in a separate file.

Please also suggest a visual abstract to illustrate your article as a PNG file 550 px wide x 300-600 px high. A cropped portion of this image will serve as thumbnail for the table of content on our webpage.

8/ As part of the EMBO Publications transparent editorial process initiative (see our Editorial at <http://embomolmed.embopress.org/content/2/9/329>), EMBO Molecular Medicine will publish online a Review Process File (RPF) to accompany accepted manuscripts.

This file will be published in conjunction with your paper and will include the anonymous referee reports, your point-by-point response and all pertinent correspondence relating to the manuscript. Let us know whether you agree with the publication of the RPF and as here, if you want to remove or not any figures from

it prior to publication.

I look forward to receiving your revised manuscript.

Yours sincerely,

Lise Roth

Lise Roth, PhD

Senior Editor

EMBO Molecular Medicine

***** Reviewer's comments *****

Referee #1 (Comments on Novelty/Model System for Author):

The manuscript has been highly improved. The authors have taken into account all my remarks and the revision is excellent!

Referee #1 (Remarks for Author):

Excellent revision !

Referee #2 (Comments on Novelty/Model System for Author):

The topic of the paper is innovative. This study has good academic value and highlights the core issues.

The authors addressed the editorial issues.

7th Apr 2025

Dear Prof. Yoon,

Thank you for submitting your revised files. I am now ready to accept your manuscript once the following remaining editorial concerns have been addressed:

1. Please remove "Reagents and Tools table" and "Methods and Protocols" in the text at the beginning of the Methods section.

2. Methods:

- o Please indicate whether the cells were authenticated and tested for mycoplasma contamination.
- o Please indicate the gender of the mice used in the experiments.
- o Please provide dilutions/concentrations used for primary antibodies.

3. Data availability: Thank you for providing the links to your datasets, please note that these must be publicly accessible before acceptance of the manuscript.

4. Figures:

- Given the high number of Appendix figures, we would strongly encourage you to make some of these figures 'Expandable View figures', that are collapsible/expandable online. EV Figures should be cited as 'Figure EV1, Figure EV2' etc... in the text and their respective legends should be included in the main text after the legends of regular figures.
- Figure S8A: similarities have been found between the top right panel (p300(low)/Adriamycin) and the bottom right panel (p300(High)/Adriamycin) (see screenshot attached). Please carefully check the composition of the figure, correct, and provide the accompanying source data.
- Figure 2B: the data for control conditions are the same for control and TGFb. Please clarify and consider a different representation.
- Source data for Figure 4D, E, F are mislabelled.

5. Checklist: In the section "Cell materials", please fill in the subsection on authentication and mycoplasma contamination. In the section "Experimental study design and statistics", please fill in the subsection on inclusion/exclusion criteria.

6. I have introduced minor modifications in your synopsis, please let me know if you agree or amend as you see fit:

"Chronic kidney disease (CKD) is typically associated with severe fibrosis, the exact pathogenesis of which remains unclear.

- Histone acetyltransferase p300 was upregulated in CKD patient kidneys and mouse renal fibrosis models.
- Proximal tubule-specific p300 knockout in mouse renal fibrosis models alleviated kidney fibrosis.
- The protein stability of p300 in proximal tubular cells was regulated by the phosphatase PPM1K.
- Proximal tubular p300 induced EndMT in adjacent endothelial cells and promotes fibrosis.
- A p300-specific inhibitor treatment mitigated kidney fibrosis in mouse renal fibrosis models."

Thank you for providing a nice visual abstract. I have cropped portion of this image to serve as thumbnail for the table of content on our webpage (attached), please let me know if you agree or provide an alternative image (115x70 pixels).

I look forward to receiving your revised manuscript.

Yours sincerely,

Lise Roth

The authors addressed the remaining editorial issues.

9th Apr 2025

Dear Prof. Yoon,

Thank you for submitting your revised files. I am pleased to inform you that your manuscript is accepted for publication and is now being sent to our publisher to be included in the next available issue of EMBO Molecular Medicine.

Yours sincerely,

Lise Roth
